# OASIS: Open Agents Social Interaction Simulations on a Large Scale

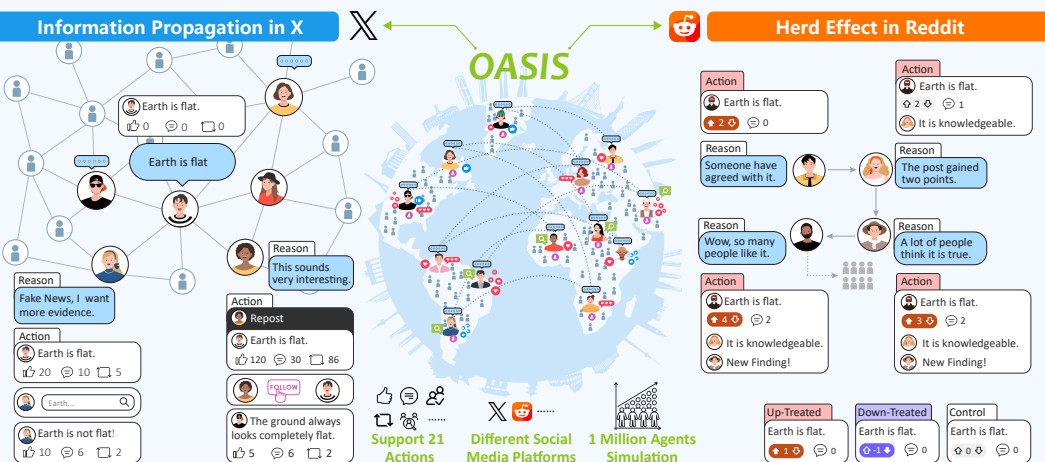

Figure 1: *OASIS* can simulate different social media platforms, such as X and Reddit, and supports simulations of up to millions of LLM-based agents.

## Abstract

There has been a growing interest in enhancing rule-based agent-based models (ABMs) for social media platforms (*i.e.*, X, Reddit) with more realistic large language model (LLM) agents, thereby allowing for a more nuanced study of complex systems. As a result, several LLM-based ABMs have been proposed in the past year. While they hold promise, each simulator is specifically designed to study a particular scenario, making it time-consuming and resource-intensive to explore other phenomena using the same ABM. Additionally, these models simulate only a limited number of agents, whereas real-world social media platforms involve millions of users. To this end, we propose *OASIS*, a generalizable and scalable social media simulator. *OASIS* is designed based on real-world social media platforms, incorporating dynamically updated environments (*i.e.*, dynamic social networks and post information), diverse action spaces (*i.e.*, following, commenting), and recommendation systems (*i.e.*, interest-based and hot-score-based). Additionally, *OASIS* supports large-scale user simulations, capable of modeling up to one million users. With these features, *OASIS* can be easily extended to different social media platforms to study large-scale group phenomena and behaviors. We replicate various social phenomena, including information spreading, group polarization, and herd effects across X and Reddit platforms. Moreover, we provide observations of social phenomena at different agent group scales. we observe that the larger agent group scale leads to more enhanced group dynamics and more diverse and helpful agents' opinions. These findings demonstrate *OASIS*'s potential as a powerful tool for studying complex systems in digital environments.

## 1 Introduction

Complex societal systems (*e.g.*, social media, cities, ecosystems, and financial markets) are characterized by many interconnected and interdependent components or agents. These interactions give rise to emergent behaviors that cannot be predicted by analyzing the actions of individual alone (Ladyman et al., 2013). These systems are important in the increasingly digital world we

live in, but conducting experiments with complex systems can be very costly in terms of time and resources. Therefore, scientists have often relied on mathematical or agent-based models (ABMs) to understand, analyze, or predict phenomena and outcomes that are difficult or impossible to conduct real-world experiments (e.g., misinformation propagation (Gausen et al., 2022), online polarization (Song & Boomgaarden, 2017), and herd effect (Lee & Lee, 2015)).

As the name suggests, ABMs consist of computational **agents** programmed to **interact** among themselves or with the **environment** in a realistic manner that is relevant to the complex system under study (Gilbert, 2019). Simulating agent behaviors is the key to designing ABMs. Traditionally, agent behaviors are programmed along measurable value (*i.e.*, thresholds), which overlooks more complex aspects such as context-dependent behavioral changes. Recently, large language models (LLMs) have demonstrated remarkable capability to mimic human behaviors (Park et al., 2022; 2023; Zhou et al., 2023b; Wang et al., 2023; Gao et al., 2023; Mou et al., 2024). LLM agents can engage in role-playing, *i.e.*, impersonating human characters and taking part in a human-like interaction with other agents (Park et al., 2023; Zhou et al., 2023b), as well as taking a wide variety of actions ranging from simple decisions to more complex ones involving the tool use (Achiam et al., 2023). To develop and evaluate these LLM agents, researchers will need to move beyond standard benchmarks by defining social situations and distinct personas, as well as integrating these agents into simulated platforms or sandbox environments for more comprehensive testing and analysis (Park et al., 2023).

In the context of social media studies, popular social media platforms (*i.e.*, X, Reddit) have drastically changed how people interact, exchange information, and form communities, making them crucial environments for studying modern social dynamics. They vary in how they design user interactions, henceforth termed *action space*, how they interact with users through algorithms and recommendation systems (*RecSys*), as well as how they connect with each other (*Dynamic Network*) For example, X facilitates a rapid exchange of views in real-time, and Reddit supports

| | # Agent | Environment | Action Space | Recsys. | Dynamic Network | LLM Support |
|---|---|---|---|---|---|---|
| Generative Agents (2023) | 25 | Town | - | × | × | OpenAI API |
| Sotopia (2023b) | 2 | - | - | × | × | OpenAI API |
| RecAgent (2023) | 5 | - | 6 | ✓ | × | OpenAI API |
| Agent4Rec (2024) | 1,000 | Movie Rec. | 5 | ✓ | × | OpenAI API |
| S3 (2023) | 1,000 | X | 4 | × | × | OpenAI API |
| HiSim (2024) | 300/700 | X | 5 | × | × | OpenAI API |
| AgentScope (2024) | **1M** | - | - | × | × | **Open-source** |
| *OASIS* (Ours) | **1M** | **X & Reddit** | **21** | ✓ | ✓ | **Open-source** |

Table 1: A comparison of LLM agent-based simulation methods is presented. # Agent represents the number of agents in the simulation. Environment refers to the environment in which the agents operate, with a '-' indicating that no specific environment has been defined. Action Space describes the types of actions supported by the simulation. Recsys. indicates whether the simulation includes recommendation systems. Dynamic Network indicates whether the simulation supports the dynamic update of user-follow networks. LLM Support specifies the primary large language model used in the simulation.

topic-based communities and emphasizes comment interaction. Consequently, users behave very differently across platforms, and as a result, several LLM-based ABM studies (see Table 1) have been proposed recently to study some aspects of social interactions on one of these platforms. Given the specific scenarios studied under these ABMs, pivoting them to study another domain remains tedious, which limits their usability to a larger social sciences community. Furthermore, these real-world social media contain millions of users. Simulating a large-scale ABM would allow for studies across multiple platforms, either individually or collectively, but it also introduces a wide range of engineering challenges. To this end, we propose *OASIS*, a collection of generalizable and scalable ABMs to simulate a wide variety of phenomena in various social media platforms.

*How OASIS works and why OASIS is generalizable?* OASIS is built upon five foundational components, as shown in Figure 2, including the Environment Server, RecSys, Agent Module, Time Engine, and Scalable Inferencer. The Environment Server is initialized using generated or real-world data. It sends agents' information, such as user descriptions and their relationships, along with posts, to the RecSys. The RecSys selects and pushes posts to agents through recommendation algorithms, determining the visibility of content for each agent. The Time Engine activates agents based on their temporal characteristics, enabling them to perform various actions such as commenting, posting, and interacting with other agents and the environment. These actions then update the environment's state in real-time. All these components can be adapted easily to experiment with different social media platforms. For instance, by adjusting specific modules, switching from one platform, such as X, to another like Reddit is possible.

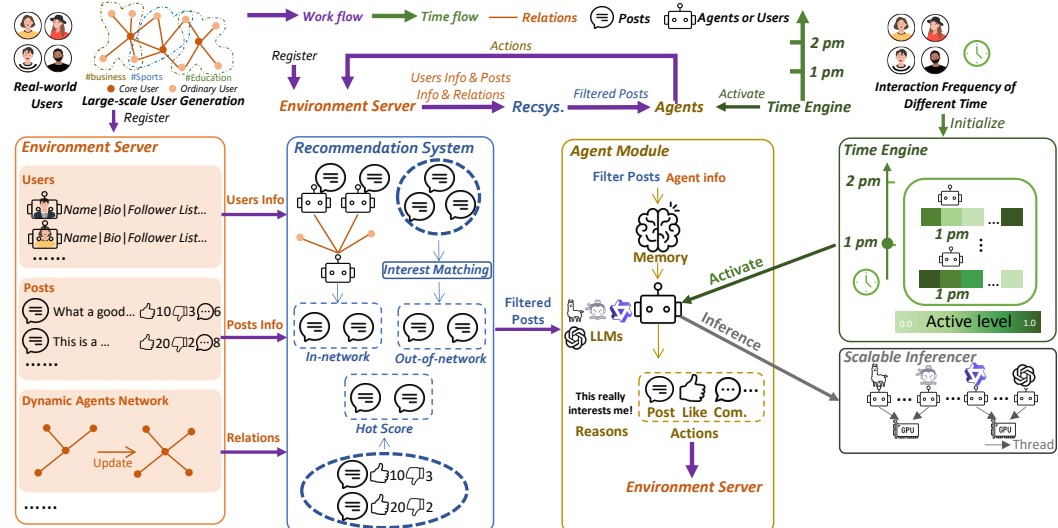

Figure 2: The workflow of *OASIS*. During the registration phase, real-world or generated user information is used to register on the Environment Server. In the simulation phase, the Environment Server sends agent information, posts, and users' relations to the RecSys, which then suggests posts to agents based on their social connections, interests, or hot score of posts. The recommended posts are then sent to the LLM-based agents, which generate actions and reasons based on the content they observe. These actions ultimately update the state of the environment in real time. The Time Engine manages the agents' temporal behaviors, while the Scalable Inferencer handles large-scale inference requests from users.

*Why scalability matters and how OASIS support scalable design?* The scale has been proven essential in domains like vision and language modeling, as certain model behaviors only emerge with sufficient scale (Kaplan et al., 2020; Zhai et al., 2022). Still, the importance of the scale of ABMs remains largely under-explored in existing literature. *OASIS* supports large-scale user simulations, ranging from hundreds to millions of agents. Our findings demonstrate that increasing the number of agents is crucial for accurately simulating group behavior and making user perspectives more valuable and diverse. To facilitate these large-scale simulations, we develop a comprehensive user generation method that enables extensive agent experiments, along with an advanced multi-processing technique to efficiently handle high-demand inference requests. Additionally, the RecSys allows agents to access information of personal interest from a large volume of data, thereby facilitating more structured and organized large-scale interactions.

To validate the effectiveness of *OASIS*, we replicate various social phenomena (such as information spreading, group polarization, and the herd effect) across different platforms (X and Reddit). The experimental results indicate that *OASIS* can closely replicate phenomena and outcomes observed in human society, including trends in information spreading, the increasing polarization of agent opinions within the interaction, and the herd effect among agents. Additionally, we also observe unique phenomena within agent societies, such as more severe group polarization in uncensored LLMs and agents being more susceptible to the herd effect compared to humans. Furthermore, we find that the number of agents plays a significant role in simulating group behavior as well as in the diversity and helpfulness of agents' opinions. We hope that *OASIS* will support research across various disciplines and contribute to the future study of agent-based societies.

## 2 METHODOLOGY

*OASIS* is developed with the aim of creating a highly generalizable LLM-based simulator for various social media. In this section, we describe the workflow as well as critical internal mechanisms of *OASIS*, which enable it to be easily generalized and scaled to support the simulation of millions of LLM-based agents.

## 2.1 WORKFLOW OF *OASIS*

*OASIS* is built upon the structure of traditional social media platforms and consists of five key components: Environment Server, RecSys, Agent Module, Time Engine, and Scalable Inferencer.

**Registration Phase.**   During the registration phase, *OASIS* requires users' information, including name, self-description, and historical posts. After registration, each user (or agent) receives a character description and an action description, guiding them to better align with their characteristics and to perform specific actions on various social media platforms.

**Simulation Phase.**   In the simulation phase, the environment sends user-related information—such as the user's past behavior and self-description to the RecSys. The RecSys filters posts from the environment and suggests posts that are likely to be of interest to the agent. Based on these posts, the agent's self-description, and other contextual factors, the agent selects actions to take, such as liking or reposting a post. Chain-of-Thought (CoT, Wei et al. (2022)) reasoning is incorporated, enabling the agent to generate reasoning alongside its actions. The agent's activation is governed by the time engine, which stores the user's hourly activity probability in a 24-dimension list. Based on these usage patterns, the time engine probabilistically activates the agent at specific times. After the agent performs actions, the results are updated in the environment server. For example, newly created posts are added to the post table in the database, or the user's relations network is updated when they follow a new user. These updates ensure that the environment accurately reflects the most recent state of the user's social network.

## 2.2 ENVIRONMENT SERVER

The role of the environment server is to maintain the status and data of social media platforms, such as users' information, posts, and user relationships. We implement the environment server using a relational database to manage and store this information efficiently. The detailed database structure is provided in the appendix C.2. The environment server is primarily composed of six components: users, posts, comments, relations, traces, and recommendations. The **user table** stores basic information about each user, such as their name and biography. The **post table** and the **comment table** each contain all the posts and comments made on the platform, including detailed information like the number of likes and the creation time. The **relations component** comprises multiple tables that store various types of relationships, such as follow and mutual relationships between users, likes between users and posts, among others. Each user's entire action history is recorded in the **trace table.** The **recommendation table** is populated by the output of the RecSys after analyzing the user's trace table. The database can be dynamically updated. For example, new users, posts, comments, and follow relationships can be added over time. This dynamic flexibility significantly enhances the versatility and usability of *OASIS*.

## 2.3 RECSYS

The role of the RecSys is to control the information seen by agents, playing a crucial part in shaping the information flow. We develop RecSys for two popular social media platforms: X and Reddit.

For X, following X official report (Twitter, 2023), the recommended posts come from two sources: in-network (users followed by the agent) and out-of-network (posts from the broader simulation world). In-network content is ranked by popularity (likes) before recommendation. Out-of-network posts, as shown in Figure 3, are recommended based on interest matching using TwHIN-BERT (Zhang et al., 2023), which models user interests based on profiles and recent activities by vectors' similarity. Factors like recency (prioritizing newer posts) and the number of followers of the post's creator (simulating superuser broadcasting) are also taken into account to recommend relevant out-of-network posts, details are presented in Appendix C.3. Additionally, the post count from in-network and out-of-network sources can be adjusted to suit different scenarios.

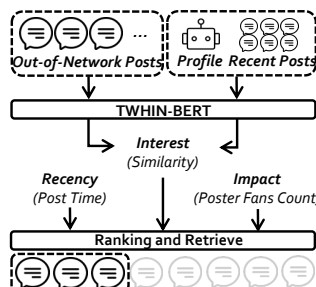

Figure 3: The pipeline of the out-of-network post recsys.

For Reddit, the RecSys is modeled based on Reddit's disclosed post ranking algorithm (Salihefendic, 2015), which calculates a hot score to prioritize posts. This score integrates likes, dislikes, and created time, ensuring that the most recent and popular posts are ranked at the top, while those less popular or controversial rank lower. Specifically, the calculation formula is:

$$h = \log_{10}\left(\max\left(|u - d|, 1\right)\right) + \text{sign}(u - d) \cdot \frac{t - t_0}{45000} \tag{1}$$

where $h$ indicates the hot score, $u$ represents the number of upvotes, $d$ represents the number of downvotes, and $t$ is the submission time in seconds since the Unix epoch, $t_0 = 1134028003$. We rank the posts based on hot scores to identify the top $k$ posts for recommendation, with the number of recommended posts (*i.e.*, $k$) varying depending on the experiment; further details are presented in Appendix E.4.2.

## 2.4 AGENT MODULE

Our agent module is based on large language models, and the core features of the agent module are inherited from CAMEL (Li et al., 2023). The agent module consists primarily of a **memory module** and an **action module**. The **memory module** stores information the agent has encountered. To help the agent better understand its role when performing actions, the memory includes sufficient information about posts, *e.g.* the number of likes, comments, and the likes on comments. Additionally, it stores the user's previous actions and the reasoning behind them. The **action module** enables 21 different types of interactions with the environment, including *sign up, refresh, trend, search posts, search users, create post, repost, follow, unfollow, mute, like, unlike, dislike, undo dislike, unmute, create comment, like comment, unlike comment, dislike comment, undo dislike comment, and do nothing*. The details of these actions are available in the Appendix C.1. We also utilize CoT reasoning to enhance the interpretability of the agent behaviors. By incorporating a larger action space, we increase user interaction diversity, making them closer to real-world social media platforms.

## 2.5 TIME ENGINE

It is crucial to incorporate temporal features into the agent's simulation to accurately reflect how their real-world identities influence online behavior patterns. To address this, we define each agent's hourly activity level based on historical interaction frequency or customized settings. Each agent is initialized with a 24-dimensional vector representing the probability of activity in each hour. The simulation environment activates agents based on these probabilities, rather than activating all agents simultaneously. Moreover, we manage time progression within the simulation environment using a time step approach (*i.e.*, one time step is equal to 3 minutes in *OASIS*), similar to the approach used in Park et al. (2023), which accommodates varying LLM inference speeds across different setups. Additionally, since the creation time of a post within a single time step is crucial for the Reddit recommendation system, we propose an alternative time-flow setting. This setting linearly maps real-world time using a scale factor to adjust the simulation time, ensuring that actions executed earlier within the same time step are recorded with earlier timestamps in the database.

## 2.6 SCALABLE DESIGN

**Scalable Inference** We design a highly concurrent distributed system where agents, the environment server, and inference services operate as independent modules, exchanging data through information communication channels. The system leverages asynchronous mechanisms to allow agents to send multiple requests concurrently, even while waiting for responses from previous interactions, and the environment module processes incoming messages in parallel. Inference services manage GPU resources through a dedicated manager, which balances agent requests across available GPUs to ensure efficient resource utilization. For more details, see Appendix C.4.

**Large-scale User Generation** The user generation algorithm addresses platform constraints and privacy concerns by combining real user data with a relationship network model, simulating up to one million users while preserving the scale-free nature of social networks. It generates diverse user profiles based on population distributions, simplifying dimensions like age, personality, and

profession as independent variables. Core and ordinary users are linked into a network using interest-based sampling, with a 0.2 probability of following core users, ensuring diversity and preventing network density. Details are presented in Appendix D.1, D.2 and D.3.

## 3 EXPERIMENT

Although *OASIS* has the potential to be applied for various computational inquiries, we primarily focus on two research questions below:

1. **Can *OASIS* be adapted to various platforms and scenarios to replicate real-world phenomena?** We demonstrate the generalizability of *OASIS* by replicating three influential computational social science studies. Specifically, we simulate information propagation (Vosoughi et al., 2018) and the resulting group polarization (Lindesmith et al., 1999) on rapid information exchange platforms like X and the herd effect (Muchnik et al., 2013) on topic-based community-oriented platforms like Reddit.
2. **Does the agent population affect the accuracy of simulating group behavior?** We conduct sociological experiments at various scales of agents, ranging from hundreds to tens of thousands of agents, and identify (if any) emergent sociological phenomena as the number of agents increases.

### 3.1 EXPERIMENTAL SCENARIOS

**Information propagation on X.**    *Information propagation* refers to the propagation of messages through a network, influenced by varied factors (*e.g.*, network structure, message content, and individual interactions). It is crucial for understanding phenomena like information spreading and group polarization. In this section, we explore two key aspects: *information spreading*, the transmission of messages across a network; and *group polarization*, where social interactions foster increasingly extreme opinions. Our analysis focuses on these dynamics within the X platform.

**Herd effect in Reddit.**    *Herd effect* refers to individuals' tendency to follow the actions or opinions of a larger group without independent thought or analysis. For example, users tend to like a post that has already received likes or reflect a general inclination to conform to majority opinions. Our analysis focuses on these dynamics within the Reddit platform.

### 3.2 EXPERIMENTAL SETTINGS

For **information spreading**, we collect 198 real-world instances from two rumor detection datasets, Twitter15 (Liu et al., 2015) and Twitter16 (Ma et al., 2016), covering 9 categories (*e.g.*, business, education, and politics). Each instance includes 100 to 700 users and information propagation path of the source post. Using the X API, we retrieve user profiles, follow relationships, and previous posts, computing users' hourly activity levels (see Appendix D.1 for details). Agents in *OASIS* are initialized with this data, and their most recent posts will also be included in the simulator to be propagated along with the source post for better alignment with real-world scenario (Section 2.1). For **group polarization**, we select 196 real users' information from the information-spreading experiment (these real users have a large following on X and they are from different areas.) and using LLMs to generate synthetic users with up to 1 million scale (Prompts and details are presented in Appendix D.2). Real users are set as core users, with generated users forming follow-up relationships based on topics like sports and entertainment. For **herd effect**, we first closely follow Muchnik et al. (2013) and collect 116,932 real comments from Reddit across seven topics and use LLMs to generate profiles for 3,600 users. Second, we collect 21,919 counterfactual content posts (Meng et al., 2022) and generate 10,000 users. Comments or posts are divided into three groups: the down-treated group (one initial dislike), the control group (no initial likes or dislikes), and the up-treated group (one initial like). We simulate 40 or 30 time steps of interactions for each experiment on Reddit, introducing initially-rated comments or posts at the beginning of each time step (Details are presented in Appendix D.3 and E.4.2). Llama3-8b-instruct is used as the base LLM. We adjust agent actions to accommodate different scenarios, with specific actions for each scenario detailed in Appendix E.1.

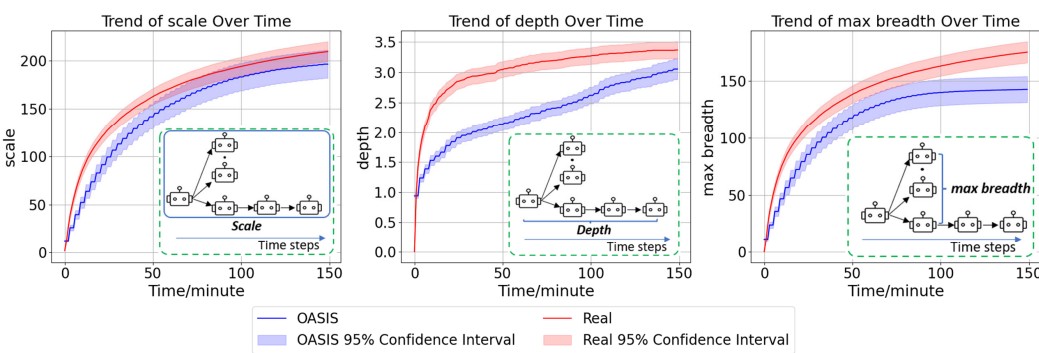

Figure 4: Mean-confidence interval distributions comparison between *OASIS* simulation results and real propagation on 198 instances. For relative magnitudes, We can observe that there is no significant offset of scale and max breadth while the depth of simulation results is noticeably lower.

**Evaluation Metrics**   For **information spreading** in X, following Vosoughi et al. (2018), we measure the information spreading paths using three key metrics: *scale* (the number of users participating in the propagation over time), *depth* (the maximum depth of the propagation graph of the source post), and *max breadth* (the largest number of users participating in the propagation at any depth). We then compute the Normalized RMSE between each simulation and real-world metric curves, averaging these values to represent *OASIS*'s overall error. Additionally, We calculate the Normalized RMSE at each minute to evaluate precise alignment and use mean and confidence intervals to understand relative magnitudes under different settings. While averaging curves makes this metric unsuitable for precise alignment with real data (For example, the error caused by a higher metric value in the simulation of source post A compared to the real data could be balanced out by a lower value in a simulation of the source post B), confidence intervals provide some level of analysis for alignment, and it helps observe relative size differences, which RMSE cannot. (For more details of these metrics please see Appendix E.2). For **group polarization**, we follow the alignment evaluation metric and the Safe RLHF Benchmark (Dai et al., 2023), using GPT-4o-mini to assess which opinions are more extreme or helpful (prompts and details are presented in Appenix E.3). This approach allows for a more precise analysis of the evolution of users' opinions. For **herd effect**, we utilize two evaluation metrics. The first is the *post score*, which is calculated as the difference between the number of upvotes and downvotes a post receives after user interaction. The second metric, the *disagree score*, is applied to counterfactual posts, where we evaluate the degree of disagreement expressed in comments responding to the counterfactual content. Further details regarding the evaluation metrics can be found in Appendix E.4.1).

### 3.3   Can *OASIS* be Adapted to Various Platforms and Scenarios to Replicate Real-world Phenomena?

#### 3.3.1   Information Propagation in X

**Finding 1: *OASIS* can replicate the information spreading process in the real world in terms of scale and maximum breadth without evident offset; however, the depth trend is smaller compared to real-world trends.** We compare the simulation information propagation process with the real-world ground truth in Figure 4. Overall, the *OASIS* simulation results align with real-world information dissemination trends well, with an error margin of normalized RMSE around 30%. This validates *OASIS*'s effectiveness in modeling these dynamics. However, we observe that the depth of *OASIS* simulation propagation is smaller than the real-world propagation in Figure 4. This discrepancy likely arises from the complexity and precision of real-world RecSys and user profiles. While our RecSys effectively captures the broadcasting effect of super users, data limitations hinder its ability to accurately represent nuanced user profiles. As a result, the simplified design of our RecSys struggles to model intermediary users with the same level of precision.

**Finding 2: *OASIS* can replicate the phenomenon of group polarization, where opinions become increasingly extreme during information propagation. This effect is even more pronounced in uncensored models.** Studying how users' opinions evolve during information propagation is crucial. Here, we examine group polarization during information propagation. *Group Polarization* oc-

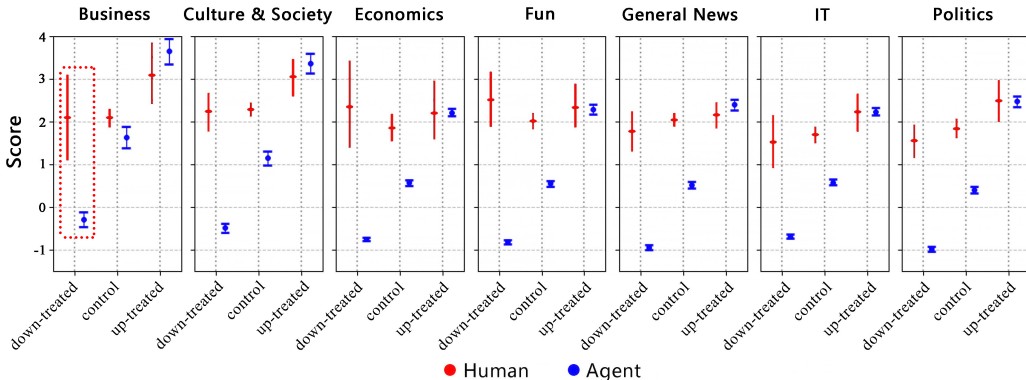

Figure 5: Evaluation results of group polarization for uncensored and aligned Llama-3-8B. The red bar indicates the opinion is more extreme compared with the round 0. The blue bar indicated more progressive and the green bar indicated draw. We also demonstrate the examples of different rounds on the right side of each figure.

Figure 6: The figure displays the mean comment scores for up-treated comments (initially liked), down-treated comments (initially disliked), and control group comments (with no likes or dislikes), along with 95% confidence intervals for both humans and LLM agents across the seven topic categories. Red indicates the results for humans, while blue represents the results for LLM agents. The red box shows that for the down-treated comments group the agents are more likely to exhibit herd effect, which differs significantly from humans.

curs when individuals with similar views adopt more extreme positions after exchanging opinions. For example, a group with moderately conservative views may become more conservative through interaction. Here, we set a hypothetical scenario where users on X discuss a classic dilemma (Linde-smith et al., 1999): *Should Halen take the risk to write a great novel, or should he continue writing ordinary novels without taking any risks?* We let one user post a discussion (see Appendix E.3.1) about the dilemma, and then the discussion was held among 196 core users. After extensive information propagation, we collect every agent's advice about *what should Halen do?* at every 10 time steps in the form of a questionnaire (see Appendix E.3.2) and analyze the changes in their views over different periods of interaction. Initially, agents are assigned conservative views with prompts. The entire simulation will last for 80 time steps, every 10 time steps we would use GPT-4o-mini to compare the opinions gathered with the initial opinions and judge which is more conservative. The results are as follows:

We discover that as the interaction progresses, agents' responses to Halen's suggestions become increasingly conservative, especially in interactions with uncensored models (The uncensored model has been stripped of its safety guardrails). The uncensored model tends to use more extreme phrases, such as 'always better' and similar expressions. These findings suggest that LLM-based agents exhibit a tendency toward extremism during social interactions, as their attitudes shift from moderate to extreme over time.

### 3.3.2 HERD EFFECT IN REDDIT

We simulate agents' interactions on comments of different topics using *OASIS* for 40 time steps. The average scores of all comments after all time steps in the experiment are shown in the figure 6.

**Finding 3: Agents are more inclined to herd effect, while humans possess a stronger critical mind.** As shown in Figure 6, for the up-treated group, the simulation results of the agent and humans are relatively close, showing a high level of consistency. However, for the down-treated group, the human group's scores are significantly higher than the results observed from agent group. This suggests that when an initial comment receives a dislike, agents tend to follow others' behavior by further disliking the post or giving fewer likes, whereas humans, on the other hand, tend to deliberate more carefully and are more likely to increase the like score.

### 3.4 DOES THE NUMBER OF AGENTS AFFECT THE ACCURACY OF SIMULATING GROUP BEHAVIOR?

#### 3.4.1 INFORMATION PROPAGATION IN X

A natural question to ask is how an increasing number of agents might influence group polarization and individual user opinions. Therefore, we conduct experiments on group polarization at different agent scales *i.e.*, 196~100K. To investigate how the same agents' opinions change across different scales, we collect suggestions from the same 196 users in all experiments. The other experimental settings are kept consistent with those described in group polarization. We run the simulation for 30 time steps. We visualize the distribution of agents' opinions at different scales using Nomic Atlas (Nomic, 2024), as shown in Figure 7.

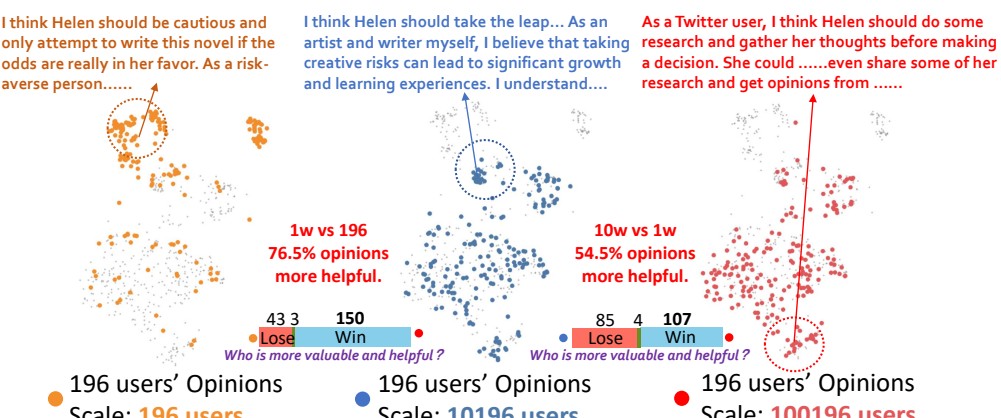

Figure 7: Visualization of 196 core users' opinions across different scale of agents and the evaluation results of helpfulness.

**Finding 4: Larger group leads to more helpful and diverse responses.** As shown in Figure 7, we find that when the number of agents increases from 196 to 10,196, there is a significant enhancement in the diversity of user opinions. Additionally, following the evaluation criteria from Safe-RLHF (Dai et al., 2023), we assess which set of user opinions—those from 196 or 10,196 agents—is more helpful. The results indicate that the helpfulness of the 10,196 agents is significantly better than that of the 196 agents. When the number of agents is further expanded to 100,196, the helpfulness of user opinions improves even more. This suggests that as the user base grows, core users are exposed to a more diverse and enriching set of responses, leading to more varied and helpful interactions.

#### 3.4.2 HERD EFFECT IN REDDIT

**Finding 5: When faced with counterfactual posts, the agent exhibits herd effect only in response to dislikes, and this effect becomes more pronounced as the number of agents increases.** In this section, we conduct an experiment to investigate whether agents would exhibit herd effect when exposed to counterfactual posts (*i.e.*, misinformation). Interestingly, we observed that when the number of agents was small, there appeared to be no herd effect, as there was no difference in scores between the up-treated, control, and down-treated groups. This raised the question of whether herd effect was truly absent. We then increased the number of agents from 100 to 10,000, and found that the agents began to exhibit explicit herd effect. The disagree scores in the down-treated group were significantly higher than those in the control and up-treated groups. Additionally, there was

a noticeable increase in the scores, suggesting that large-scale groups tend to guide agents toward self-correction. For specific examples of this phenomenon, illustrated through posts and comments, see Appendix E.4.3.

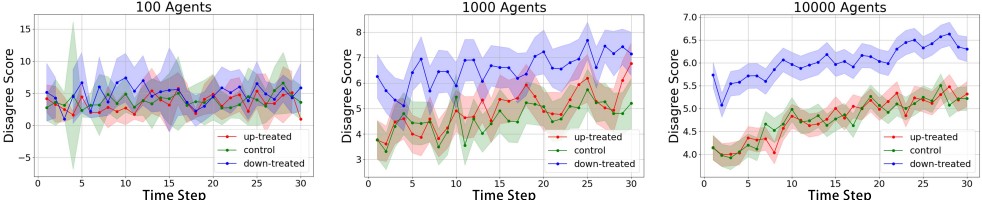

Figure 8: The disagree scores of agents' comments created at all time steps and across different scales of agents. The red, blue, and green curves represent the up-treated, down-treated, and control groups, respectively. We present the mean and the 95% confidence intervals for all results.

## 4 ABLATION STUDY

### 4.1 ANALYSIS OF EFFICIENCY FOR MILLIONS OF USERS

In this study, we report the runtime and GPU utilization for simulations at scales of one million, one hundred thousand, and ten thousand under a group polarization setting, as well as the number of tweets and comments added at each time step. For all scenarios, we use one A100 for RecSys and use multiple GPUs for LLM inference. We use vLLM (Kwon et al., 2023) to efficiently conduct LLM inference. As shown in Table 2, our algorithm can efficiently simulate large-scale user interactions. For in-

| Scale | 1M | 100K | 10K |
|---|---|---|---|
| Hours per time step | 18.0 | 3.0 | 0.2 |
| GPUs (A100) | 27.0 | 5.0 | 2.0 |
| New Tweets per time step (K) | 48.5 | 5.2 | 0.6 |
| New Comments per time step (K) | 97.1 | 9.0 | 0.9 |

Table 2: Experiment efficiency analysis of different agent scale. K stands for 1000. M stands for one million

stance, using five A100 GPUs, we can simulate the interactions of 100,000 users over 10 time steps within two days. Other scenarios' efficiency analysis are presented in Appendix B.1.

### 4.2 ABLATION OF COMPONENTS IN *OASIS*

We conduct ablation experiments on various modules of *OASIS*, including the RecSys, and the temporal feature used in Time Engine. For the RecSys, we find that its absence significantly hampers the spread of information, limiting the potential for wide dissemination. Testing different models such as MiniLM v6 (Reimers & Gurevych, 2019), BERT (Devlin, 2018), and TwHIN-BERT. We observe that TwHIN-BERT, which pre-trained on over 7 billion tweets in 100+ languages, performs particularly well in capturing similarities between different posts. For the temporal feature, we replace the 24-dimensional activity probability list, extracted from the crawled user's previous post frequency, with a list where each dimension is set to 1. The results demonstrate that the activity probability from real-world data is essential for accurately reproducing real-world data dissemination patterns. Further visualization and experiment results can be found in Appendix B, The primary metric we use here is the Normalized RMSE at every minute for a more detailed analysis.

## 5 CONCLUSION

We present *OASIS*, a generalizable and scalable social media simulator designed to replicate real-world social media dynamics. *OASIS* incorporates modular components that capture the core functionalities of social media platforms, enabling it to be easily adapted across different platforms. Moreover, *OASIS* supports large-scale user interactions, accommodating up to 1 million users. Using *OASIS*, we have reproduced several well-known social phenomena and uncovered unique behaviors emerging from LLM-driven simulations. We also identified distinctive patterns in group behavior that vary with different group sizes. We hope *OASIS* can provide valuable insights for future research on social group dynamics and general multi-agent interactions.

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

## A  RELATED WORK

### A.1  SOCIAL MEDIA

Social media encompasses websites and applications focused on communication, interaction, and content-sharing (Kapoor et al., 2018). While it offers benefits like allowing individuals to explore their identities without real-world consequences (Nature Reviews Psychology, 2024), the risk of hazardous social media phenomena gradually becomes a global threat with significant economic, political, and social consequences. Traditional threats includes promoting risky behaviors (Nature Reviews Psychology, 2024), contributing to mental health issues among teenagers (Odgers, 2024), social influence (Muchnik et al., 2013), group Polarization (Iandoli et al., 2021; Isenberg, 1986), and spreading misinformation (Vosoughi et al., 2018; Waldrop, 2023). Despite numerous studies on social media phenomena, the complex network structures, vast data, and diverse behaviors present challenges for researchers. Additionally, ethical concerns (Moreno et al., 2013) arise in some of these studies. To address these issues, a controllable virtual environment (*e.g.*, a multi-agent system) for social simulation is needed, allowing researchers to test hypotheses on a virtual platform.

### A.2  MULTI-AGENT SYSTEMS

Multi-agent systems are composed of multiple autonomous entities, each possessing different information and diverging interests. Compared to single-agent platforms, multi-agent platforms offer several advantages, including (1) the ability to assume different roles in group activities, and (2) richer and more complex interaction behaviors, such as collaboration, discussion, and strategic competition. Recent studies have demonstrated the potential of multi-agent systems across various domains. Divided by various functionality, recent multi-agent systems can be roughly divided to tool-based agent assistants (Qian et al., 2023; Zhao et al., 2024; Mosquera et al., 2024; Wang et al., 2024), as well as society or game simulation environments (Li et al., 2023; Zhou et al., 2023a; Huang et al., 2024; Yu et al., 2024). The former part focus on collaborating a small group of LLM-based agents to automatically conduct predefined or open-ended tasks. And the latter part focus on involving a large-scale agent groups to automatically run a simulator in a specific environment. Since the action and relationship in a large society is extremely complicated, capability scalability has become the fundamental issue of this work. In this work, we highly focus on leveraging multi-agent systems to explore corresponding characteristics in social simulation research.

### A.3  MULTI-AGENT SYSTEM SOCIAL SIMULATION

Social simulation plays a crucial role in social science research, with many classic agent-based modeling (ABM) studies, such as Schelling's model of segregation (Schelling, 1969) and the Chicago simulation (Macal et al., 2018). Traditional ABM has limitations such as subjective rule design and scalability issues. With the development of large language models (LLMs), LLM-based agents have demonstrated significant advantages in social simulation: (1) The ability to interact using natural language. (2) A more accurate simulation of human behavior. (3) The capability to utilize more complex tools. There have been numerous related studies, such as the exploration of multi-agent behavior patterns (Park et al., 2023), simulations of social networks (Gao et al., 2023; Zhou et al., 2023b), and the study of society's response to misinformation (Chen & Shu, 2023). Social simulation not only serves as a tool for social science research but also aids in exploring the boundaries of LLMs' capabilities. For example, studies on social alignment (Liu et al., 2023), emergence of social norms (Ren et al., 2024). However, current LLM-related social simulations mainly focus on interactions among a small number of agents. Yet, research on collective behavior often requires a critical mass to observe emergent phenomena. Therefore, our work emphasizes the interaction of large-scale agents to study the emergence of collective behaviors.

## B  ABLATION STUDY

### B.1  MORE EFFICIENCY ANALYSIS

Table 3 presents the efficiency analysis of the Counterfactual herd effect experiment 3.4.2 in Reddit.

Table 3: Experiment efficiency analysis of different agent scales.

| Scale | 10k | 1k | 100 |
|---|---|---|---|
| Minutes per time step | 15 | 0.83 | 0.33 |
| GPUs (A100) | 4 | 4 | 4 |
| New Comments per time step | 1393 | 129 | 14 |

## B.2 RECOMMEND SYSTEM ABLATION

To verify the impact of RecSys on message dissemination, we conduct ablation studies on the existence of the RecSys itself and the RecSys model (different models to embed posts and profiles). For these experiments, we randomly select 28 topics (Here, 'topic' refers to a propagation instance, with more emphasis on the topic type of the source post.) from the 198 topics collected before, ensuring that they still cover 9 categories.

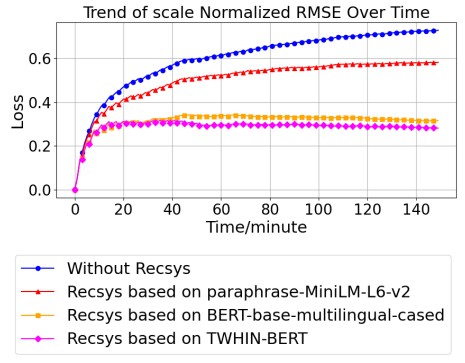

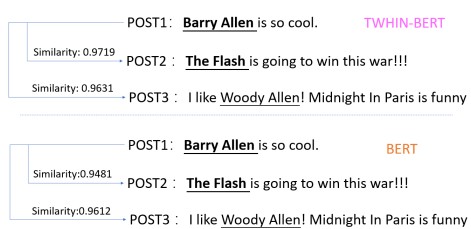

(b) Recommendation results of TwHIN-BERT and regular BERT. TwHIN-BERT can identify the relationship between Barry Allen and The Flash (Barry Allen is the second-generation Flash), whereas regular BERT would not be able to achieve this.

(a) RecSys ablation results on scale Normalized RMSE, TwHIN-BERT and regular BERT show much better performance.

Figure 9: Recsys ablation results and recommendation results comparison.

**w/o RecSys.** In our experiments, removing the RecSys for some entertainment topics worked well due to dense follower networks in fan groups. However, most groups lack these networks, and removing the RecSys leads to the premature end of information spread, typically manifesting as broadcast behavior from a single superuser. Thus, the RecSys is essential for connecting isolated nodes and sustaining the simulation.

**Different RecSys model.** Pre-trained on over 7 billion posts in 100+ languages, TwHIN-BERT is more suitable for recommendation systems than general models. Here we choose paraphrase-MiniLM-L6-v2 and BERT-base-multilingual-cased (regular BERT) for the ablation study, we found that TWHIN-BERT and regular BERT show much better performance than paraphrase-MiniLM-L6-v2 in Figure 9a. Moreover, based on recommendation results in Figure 9b, TWHIN-BERT could recommend a more proper post.

## B.3 TEMPORAL FEATURE ABLATION

We ablate our temporal feature (the hourly activity level extracted from the crawled data) in this experiment. Specifically, we rerun the experiments of reproducing real-world information propagation under all activity probabilities set to 1.0 and compare their Normalized RMSE on 28 topics. We can easily see that without the temporal features, our *OASIS* can not capture the dynamics of real-world information propagation well since all agents take action so frequently.

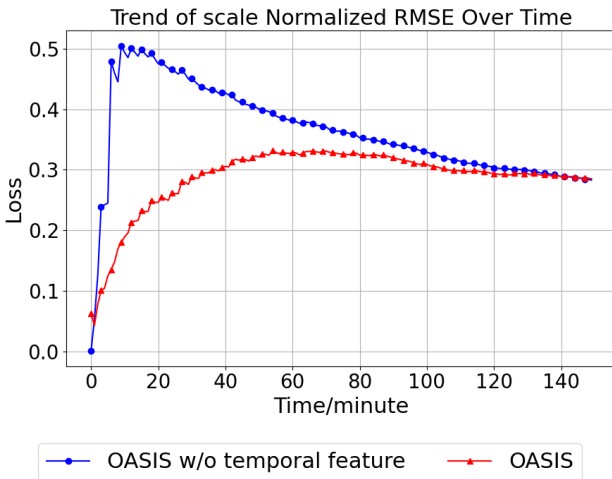

Figure 10: Normalized RMSE between *OASIS*, *OASIS* w/o temporal feature simulation results and real propagation.

### B.4 LLM ABLATION

We tried different open-sourced LLMs including Qwen1.5-7B-Chat, Internlm2-chat-20b, and Llama-3-8B-Instruct as the backend of agents on the experiments of reproducing real-world information propagation (still on 28 topics randomly picked before).

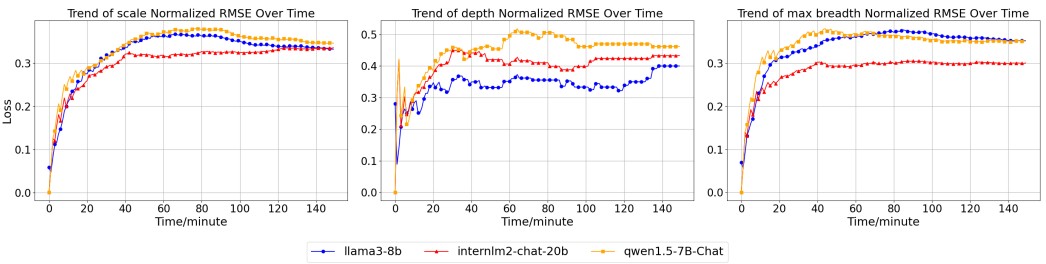

Figure 11: Normalized RMSE of simulation results of different LLM-based agents.

## C METHOD DETAILS

### C.1 USER ACTIONS PROMPTS

Note: This section outlines the complete set of 21 actions available within the action space. However, for our different experiments, we flexibly select a subset of these actions based on the specific requirements of each study.

```
# OBJECTIVE
You're a Twitter/Reddit user, and I'll present you with some posts
    . After you see the posts, choose some actions from the
    following functions.

- sign_up: Signs up a new user with the provided username, name,
    and bio.
    - Arguments:
        "user_name" (str): The username for the new user.
        "name" (str): The full name of the new user.
        "bio" (str): A brief biography of the new user.
```

- create_post: Create a new post with the given content.
  - Arguments: "content" (str): The content of the post to be created.
- repost: Repost a post.
  - Arguments: "post_id" (integer) – The ID of the post to be reposted. You can `repost` when you want to spread it.
- like_post: Likes a specified post.
  - Arguments: "post_id" (integer) – The ID of the post to be liked. You can `like` when you feel something interesting or you agree with.
- unlike_post: Removes a previous like from a post.
  - Arguments: "post_id" (int): The ID of the post from which to remove the like. You can `unlike` when you reconsider your stance or if the like was made unintentionally.
- dislike_post: Dislikes a specified post.
  - Arguments: "post_id" (integer) – The ID of the post to be disliked. You can use `dislike` when you disagree with a post or find it uninteresting.
- undo_dislike_post: Removes a previous dislike from a post.
  - Arguments: "post_id" (int): The ID of the post from which to remove the dislike. You can `undo_dislike` when you change your mind or if the dislike was made by mistake.
- create_comment: Creates a comment on a specified post to engage in conversations or share your thoughts on a post.
  - Arguments:
    "post_id" (integer) – The ID of the post to comment on.
    "content" (str) – The content of the comment.
- like_comment: Likes a specified comment.
  - Arguments: "comment_id" (integer) – The ID of the comment to be liked. Use `like_comment` to show agreement or appreciation for a comment.
- unlike_comment: Removes a previous like from a comment.
  - Arguments: "comment_id" (integer) – The ID of the comment from which to remove the like. Use `unlike_comment` when you change your opinion about the comment or if the like was made by accident.
- dislike_comment: Dislikes a specified comment.
  - Arguments: "comment_id" (integer) – The ID of the comment to be disliked. Use `dislike_comment` when you disagree with a comment or find it unhelpful.
- undo_dislike_comment: Removes a previous dislike from a comment.
  - Arguments: "comment_id" (integer) – The ID of the comment from which to remove the dislike. Use `undo_dislike_comment` when you reconsider your initial reaction or if the dislike was made unintentionally.
- follow: Follow a user specified by 'followee_id'. You can `follow` when you respect someone, love someone, or care about someone.
  - Arguments: "followee_id" (integer) – The ID of the user to be followed.
- unfollow: Stops following a user.
  - Arguments:
    "followee_id" (int): The user ID of the user to stop following.
- mute: Mute a user specified by 'mutee_id'. You can `mute` when you hate someone, dislike someone, or disagree with someone.
  - Arguments: "mutee_id" (integer) – The ID of the user to be muted.

```
- unmute: Unmute a user specified by 'mutee_id'. You can unmute
   when you decide to stop ignoring their content or wish to see
   their messages and posts again.
   - Arguments: "mutee_id" (integer) - The ID of the user to be
      unmuted.
- search_posts: Searches for posts based on specified criteria.
   - Arguments: "query" (str) - The search query to find relevant
      posts. Use `search_posts` to explore posts related to
      specific topics or hashtags.
- search_user: Searches for a user based on specified criteria.
   - Arguments: "query" (str) - The search query to find relevant
      users. Use `search_user` to find profiles of interest or to
       explore their posts.
- trend: Retrieves the current trending topics.
   - No arguments required. Use `trend` to stay updated with what'
      s currently popular or being widely discussed on the
      platform.
- refresh: Refreshes the feed to get the latest posts.
   - No arguments required. Use `refresh` to update your feed with
       the most recent posts
- do_nothing: Most of the time, you just don't feel like reposting
    or liking a post, and you just want to look at it. In such
   cases, choose this action "do_nothing"

# SELF-DESCRIPTION
Your actions should be consistent with your self-description and
   personality.

{description}

# RESPONSE FORMAT
Your answer should follow the response format:

{{
   "reason": "your feeling about these posts and users, then
      choose some functions based on the feeling. Reasons and
      explanations can only appear here.",
   "functions": [{{
      "name": "Function name 1",
      "arguments": {{
         "argument_1": "Function argument",
         "argument_2": "Function argument"
      }}
   }}, {{
      "name": "Function name 2",
      "arguments": {{
         "argument_1": "Function argument",
         "argument_2": "Function argument"
      }}
   }}]  }})
}}

Ensure that your output can be directly converted into **JSON
   format**, and avoid outputting anything unnecessary! Don't
   forget the key `name`.
```

## C.2 Environment Server Database Structure

In this section, we showcase all tables and provide examples of the data contained within the database below.

Table 4: Post table

| post_id | user_id | content | created_at | num_likes | num_dislikes |
|---|---|---|---|---|---|
| 1 | 1 | "I want to share my view by creating a post." | 2024-08-04 08:12:00 | 1 | 1 |
| ... | ... | ... | ... | ... | ... |

Table 5: Dislike table

| dislike_id | user_id | post_id | created_at |
|---|---|---|---|
| 1 | 3 | 1 | 2024-08-04 23:40:03 |
| ... | ... | ... | ... |

Table 6: Like table

| like_id | user_id | post_id | created_at |
|---|---|---|---|
| 1 | 2 | 1 | 2024-08-05 10:05:23 |
| ... | ... | ... | ... |

Table 7: Comment table

| comment_id | post_id | user_id | content | created_at |
|---|---|---|---|---|
| 1 | 1 | 2 | I agree with the post! | 2024-08-05 10:05:23 |
| ... | ... | ... | ... | ... |

Table 8: Comment Dislike table

| comment_dislike_id | user_id | comment_id | created_at |
|---|---|---|---|
| 1 | 2 | 1 | 2024-08-06 11:45:03 |
| ... | ... | ... | ... |

Table 9: Comment Like table

| comment_like_id | user_id | comment_id | created_at |
|---|---|---|---|
| 1 | 3 | 1 | 2024-08-06 12:22:30 |
| ... | ... | ... | ... |

Table 10: User table

| user_id | agent_id | user_name | name | bio | created_at | num_followings | num_followers |
|---|---|---|---|---|---|---|---|
| 1 | 1 | alice0101 | Alice | Passionate about law... | 2024-08-03 10:05:23 | 0 | 0 |
| 2 | 2 | bob_good | Bob | Hospitality enthusiast — ISTJ... | 2024-08-03 11:15:33 | 0 | 1 |
| 3 | 3 | cindy_infp | Cindy | INFP — Business Management... | 2024-08-03 12:03:02 | 1 | 0 |
| ... | ... | ... | ... | ... | ... | ... | ... |

Table 11: Follow table

| follow_id | follower_id | followee_id | created_at |
|---|---|---|---|
| 1 | 3 | 2 | 2024-08-07 13:20:34 |
| ... | ... | ... | ... |

Table 12: Mute table

| mute_id | muter_id | mutee_id | created_at |
|---|---|---|---|
| 1 | 2 | 1 | 2024-08-07 10:10:24 |
| ... | ... | ... | ... |

Table 13: Trace table

| user_id | created_at | action | info |
|---|---|---|---|
| 1 | 2024-08-03 10:05:23 | sign_up | {"name": "Alice", "user_name": "alice0101", "bio": "..."} |
| 2 | 2024-08-03 11:15:33 | sign_up | {"name": "Bob", "user_name": "bob_good", "bio": "..."} |
| 3 | 2024-08-03 12:03:02 | sign_up | {"name": "Cindy", "user_name": "cindy_infp", "bio": "..."} |
| 1 | 2024-08-04 08:12:00 | create_post | {"content": "I want to share my view by creating a post."} |
| 3 | 2024-08-04 23:40:03 | dislike_post | {"post_id": 1} |
| 2 | 2024-08-05 10:05:23 | like_post | {"post_id": 1} |
| 2 | 2024-08-05 10:05:23 | create_comment | {"post_id": 1, content": "I agree with the post!"} |
| 2 | 2024-08-06 11:45:03 | like_comment | {"comment_id": 1} |
| 3 | 2024-08-06 12:22:30 | dislike_comment | {"comment_id": 1} |
| 3 | 2024-08-07 10:10:24 | mute | {"user_id": 1} |
| 2 | 2024-08-07 13:20:34 | follow | {"user_id": 1} |
| ... | ... | ... | ... |

Table 14: Rec table (recommendation system cache)

| user_id | post_id |
|---------|---------|
| 1       | 2       |
| 2       | 2       |
| 2       | 4       |
| 3       | 1       |
| ...     | ...     |

## C.3 RECOMMENDATION SYSTEM

The recommendation system ranks all posts and saves the highest-ranked ones in a recommendation table within the database. The size of this table can be adjusted, though it remains the same for all users during a given experiment.

When an agent selects the refresh action, the environment server retrieves the post IDs linked to the user's ID from the recommendation table. A subset of these post IDs is then randomly sampled, and the environment server queries the post table to retrieve the full content of the corresponding posts, which are then sent to the user.

The recommendation algorithm used in X can be summarized by the following formula, which calculates the score between a post and a user.

$$\text{Score} = R \times F \times S \tag{2}$$

where:

$$R = \ln\left(\frac{271.8 - (t_{\text{current}} - t_{\text{created}})}{100}\right) \tag{3}$$

$$F = \max\left(1, \log_{1000}(\text{fan count} + 1)\right) \tag{4}$$

$$S = \text{cosine similarity}\,(E_p, E_u) \tag{5}$$

In this context:

- $R$ refers to the recency score.
- $t_{\text{current}}$ represents the current timestamp.
- $t_{\text{created}}$ refers to the timestamp when the post was created.
- $F$ refers to the fan count score.
- $E_p$ is the embedding of the post content.
- $E_u$ is the embedding of the user profile and recent post content.
- $S$ refers to the cosine similarity between the embeddings $E_p$ and $E_u$.

## C.4 PARALLEL OPTIMIZATION

**Information Channel**: During social simulations, multiple agents asynchronously and concurrently interact with both the social media environment and the inference management servers. To facilitate this, the server utilizes an advanced event-driven architecture that broadens event categories to encompass various agent actions and large model inference requests. Communications between the agents and the servers are facilitated through a dedicated channel. This channel comprises an asynchronous message queue to receive agent requests and a thread-safe dictionary for response storage. Upon receiving a request message from an agent, the information channel automatically assigns a UUID to ensure traceability. After processing the request, the server stores the response in the dictionary, using the UUID as the key. See Fig.12.

**Inference Manager**: The manager within the inference service is capable of managing GPU devices. This enables our system to flexibly scale the number of graphics cards up or down. Additionally, the manager can distribute inference requests from agents as evenly as possible across all graphics cards for processing, thereby ensuring the efficient utilization of GPU resources.

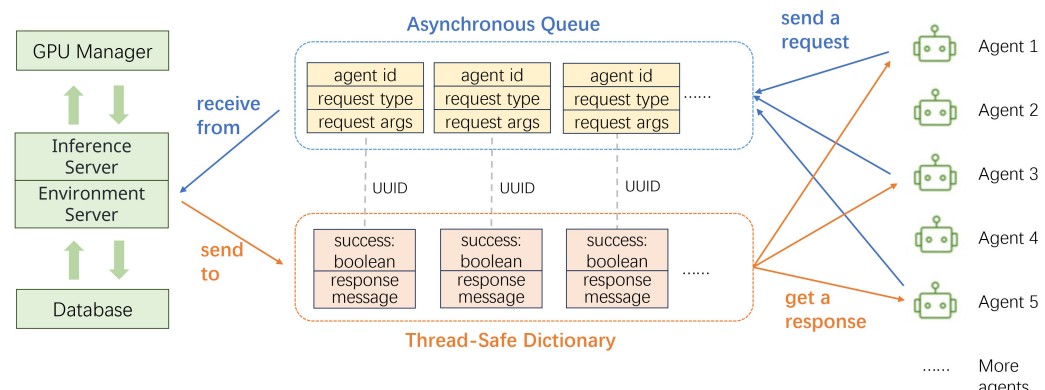

Figure 12: Architecture of information channel.

## D  DATA PREPARATIONS

### D.1  REAL-WORLD PROPAGATION DATA

We randomly select 198 propagations from Liu et al. (2015) and Ma et al. (2016), Each propagation dataset provides the source post's posting time, post content, and the propagation tree, with each node containing the user ID, repost ID, and repost time. We first use the user IDs from the propagation tree to retrieve the corresponding user's profile, the following list, and previous posts. The time period for retrieving previous posts is set to three days before the source post's posting. It is important to note that due to the high cost of data collection, we only collect posts from specific time periods within these three days, such as the hour before the source post's posting and the two hours following the source post's posting each day. Posts from the hour before the source post's posting are included in the simulation as extra noise to simulate real-world conditions better. Furthermore, since user profiles contain only basic descriptions, we would prompt GPT-3.5 Turbo to generate more detailed user profiles based on the user profiles and all previous posts. The recommendation system would use this detailed profile to create a richer user representation. The prompt template is as follows:

```
Generate a character description based on the following user
    information:
- Name: {name}
- Username: {username}
- Description: {description}
- Account Created: {created_at}
- Followers Count: {followers_count}
- Following Count: {following_count}
- Sample of Previous Posts: {previous_posts}

Please include inferred personality traits and a summary of their
    Twitter activity. Only return a short description.
```

Additionally, each user's hourly activity probability within 24 hours is calculated by the following formula:

$$P_{ij} = \frac{f_{ij}}{\max_k(f_{kj})} \tag{6}$$

The jth hourly activity probability of user i, $P_{ij}$, is calculated by the jth hourly activity frequency of user i, $f_{ij}$, divided by the maximum jth hourly activity frequency across all users in the group, $\max_k(f_{kj})$.

## D.2 Group Polarization

In this section, we provide a detailed explanation of the principles underlying the user generation algorithm. Due to platform constraints and the need to protect user privacy, large-scale scraping of user data is impractical. Moreover, conventional data scraping methods fail to guarantee a realistic relationship network, which could compromise the accuracy of propagation studies. To address these challenges, we employ a relationship network generation algorithm that combines a small amount of real user data to create a social network of up to one million users, while preserving the scale-free nature of social networks (Barabási & Albert, 1999). In this context, the user generation algorithm is the foundational data source for large-scale interactions. Our algorithm generates diverse user profiles based on real distribution data and constructs social networks based on user interests. Specifically:

**User Profiles.** To ensure the group's diversity, we acquire population distributions from disclosed statistics on social networks, including age and personality traits (in this experiment, we use MBTI as a proxy). Based on authoritative statistical data, we classify professions into 13 categories and social network trends into 9 categories, with specific categories and definitions detailed in the appendix. While ensuring scientific accuracy and diversity, we simplify the generation costs by approximating dimensions such as age, personality, and profession as independent and identically distributed random variables. We sample from these distributions, and the large model generates the agents' backgrounds and social characteristics based on this information. The prompt is as follows:

```
 Please generate a social media user profile based on the provided
     personal information, including a realname, username, user
    bio, and a new user persona. The focus should be on creating a
     fictional background story and detailed interests based on
    their hobbies and profession.
Input:
   age: {age}
   gender: {gender}
   mbti: {mbti}
   profession: {profession}
   interested topics: {topics}
Output:
{{
   "realname": str, realname,
   "username": str, username,
   "bio": str, bio,
   "persona": str, user persona,
}}
Ensure the output can be directly parsed to **JSON**, do not
    output anything else.
```

**Social Network.** Linking the large-scale generated agents into a relationship network is essential. The Matthew effect observed on social platforms distinguishes core users from ordinary users; core users on X, defined as those with more than 1000 followers, account for 80% of all users (Wojcieszak et al., 2022). Based on this, we derive an initial core-ordinary user attention tree from core users within specific interest areas, thereby constructing the initial relationship network. Specifically, each agent samples twice from an independent and identically distributed interest category distribution to obtain two topics of interest. If a topic aligns with a core user, the agent has a probability of following that core user. To prevent an excessively dense relationship network and enhance the diversity of information visible to various users, we establish the following probability at 0.1.

## D.3 Herd Effect

**User Generation.** In our Reddit experiment, the process of generating users is divided into three main steps. Initially, we reference the actual demographic distribution of Reddit users (Duarte, 2024), assigning demographic information such as MBTI, age, gender, country, and profession to each user through random sampling. Subsequently, we employ GPT-3.5 Turbo to select topics of

potential interest to the users based on the aforementioned information, choosing from seven categories: Business, Culture & Society, Economics, Fun, General News, IT, and Politics. Finally, using demographic information and selected topics, GPT-3.5 Turbo is utilized to generate each user's real name, username, bio, and persona. The generation prompts for the second and third parts are as follows.

```
# Prompt of Step-2
Based on the provided personality traits, age, gender and
    profession, please select 2-3 topics of interest from the
    given list.
    Input:
        Personality Traits: {mbti}
        Age: {age}
        Gender: {gender}
        Country: {country}
        Profession: {profession}
    Available Topics:
        1. Economics: The study and management of production,
            distribution, and consumption of goods and services.
            Economics focuses on how individuals, businesses,
            governments, and nations make choices about allocating
            resources to satisfy their wants and needs, and tries to
             determine how these groups should organize and
            coordinate efforts to achieve maximum output.
        2. IT (Information Technology): The use of computers,
            networking, and other physical devices, infrastructure,
            and processes to create, process, store, secure, and
            exchange all forms of electronic data. IT is commonly
            used within the context of business operations as
            opposed to personal or entertainment technologies.
        3. Culture & Society: The way of life for an entire society,
             including codes of manners, dress, language, religion,
            rituals, norms of behavior, and systems of belief. This
            topic explores how cultural expressions and societal
            structures influence human behavior, relationships, and
            social norms.
        4. General News: A broad category that includes current
            events, happenings, and trends across a wide range of
            areas such as politics, business, science, technology,
            and entertainment. General news provides a comprehensive
             overview of the latest developments affecting the world
             at large.
        5. Politics: The activities associated with the governance
            of a country or other area, especially the debate or
            conflict among individuals or parties having or hoping
            to achieve power. Politics is often a battle over
            control of resources, policy decisions, and the
            direction of societal norms.
        6. Business: The practice of making one's living through
            commerce, trade, or services. This topic encompasses the
             entrepreneurial, managerial, and administrative
            processes involved in starting, managing, and growing a
            business entity.
        7. Fun: Activities or ideas that are light-hearted or
            amusing. This topic covers a wide range of entertainment
             choices and leisure activities that bring joy, laughter
            , and enjoyment to individuals and groups.
    Output:
    [list of topic numbers]
```

```
Ensure your output could be parsed to **list**, don't output
    anything else.

# Prompt of Step-3
Please generate a social media user profile based on the provided
    personal information, including a real name, username, user
    bio, and a new user persona. The focus should be on creating a
     fictional background story and detailed interests based on
    their hobbies and profession.
    Input:
      age: {age}
      gender: {gender}
      mbti: {mbti}
      profession: {profession}
      interested topics: {topics}
    Output:
    {{
      "realname": "str",
      "username": "str",
      "bio": "str",
      "persona": "str"
    }}
    Ensure the output can be directly parsed to **JSON**, do not
      output anything else.
```

**Posts and Comments Dataset** In Experiment 3.3.2, we utilize a dataset comprising authentic Reddit comments and llm-generated posts. In Experiment 3.4.2, we employ a counterfactual dataset to simulate posts.

- **Real Data**: To align with human experiment Muchnik et al. (2013), our dataset included real comments and post titles from 17 subreddits during March 2023 on Reddit (Pushshift, 2023). We generate contextually relevant post content based on these titles and comments. The prompt used for generation is as follows.

```
Please generate a contextual and smooth post for this comment
and notice that the comments are correct: '{comment}'. The
response should be approximately 300 characters long and
provide relevant information or analysis. Be careful to
output the content of the post directly, and be aware that
you don't see comments when you post. And you don't need to
prefix something like: 'Here is your generated post:\n\n\'
```

Subsequently, we categorized the content from different subreddits into seven topics—Business, Culture & Society, Economics, Fun, General News, IT, and Politics—to match the categories used in human experiments. In total, we collected 116,932 comments. The specifics are detailed in the table 15.

Table 15: Details of real Reddit comments and generated posts by topic.

| Subreddit | Topic | Numbers of Posts | Numbers of Comments |
|---|---|---|---|
| Economics
finance
personalfinance | **Economics** | 4231 | 21650 |
| it
InformationTechnology
technology
learnprogramming | **IT** | 4020 | 18622 |
| AskHistorians
AskAnthropology
worldbuilding | **Culture & Society** | 2319 | 10489 |
| worldnews | **news** | 2874 | 19134 |
| politics
NeutralPolitics | **politics** | 2690 | 21477 |
| business
smallbusiness | **business** | 1807 | 8043 |
| fun | **fun** | 3272 | 17517 |

- **Counterfactual Data**: We utilize all counterfactual information from the dataset (Meng et al., 2022), comprising 21,919 entries, to create content for posts. Some examples are shown in the table 16.

Table 16: Examples of counterfactual posts.

| Counterfactual Posts |
|---|
| Shanghai is a twin city of Atlanta |
| The location of Battle of France is Seattle |
| Michel Denisot spoke the language Russian |
| The mother tongue of Go Hyeon-jeong is French |

# E    EXPERIMENTS DETAILS

## E.1    ACTIONS OF DIFFERENT SCENARIOS

Due to the significant variations between different scenarios and platforms, we adjust the agents' actions accordingly. These actions are integrated into the *OASIS* framework, allowing users to freely select and combine them. The actions for different scenarios are outlined in Table 17.

## E.2    INFORMATION SPREADING

### E.2.1    METRICS

We measure the propagation trends of messages using three key metrics: scale, depth, and max breadth. Below is a clear definition of each measure:

- **Scale**: The scale of propagation corresponds to the number of unique users involved, as each user can only repost a post once on X.
- **Depth**: A node's depth is determined by the number of edges connecting it to the root node (the original post). The overall depth of propagation is the greatest depth among all the nodes involved.
- **Max Breadth**: The breadth of propagation depends on its depth, with the number of nodes at each level representing the breadth at that specific depth. The maximum breadth is the highest number of nodes found at any depth throughout the entire propagation.

Table 17: Action type comparison across Scenarios.

| Action Type | | | | |
|---|---|---|---|---|
| **Information Spreading in X** | | | | |
| like post | repost | follow | do nothing | |
| **Group Polarization in X** | | | | |
| do nothing | repost | like post | dislike post | follow |
| create comment | like comment | dislike comment | | |
| **Comparison with the Herd Effect in Humans** | | | | |
| like comment | dislike comment | like post | dislike post | search posts |
| search users | trend | refresh | do nothing | |
| **Counterfactual Herd Effect in Reddit** | | | | |
| create comment | like comment | dislike comment | like post | dislike post |
| search users | trend | refresh | do nothing | |

Besides, the Normalized RMSE is computed as the following formula:

$$\text{Normalized RMSE} = \frac{\sqrt{\frac{1}{n}\sum_{i=1}^{n}\left(y_{\text{simu}}^{i} - y_{\text{real}}^{i}\right)^2}}{y_{\text{real}}^{n}} \tag{7}$$

Let $n$ refer to the maximum minute in the simulation results, and $y_{\text{simu}}^{i}$, $y_{\text{simu}}^{i}$ represents the value of a certain metric at the $i$th minute of the simulation process or the real-world propagation process. For Normalized RMSE at every minute, since we only compute the discrepancy between the two data points of simulation result and real propagation, the error of $i$-th minute can be calculated by $|y_{\text{simu}}^{i} - y_{\text{real}}^{i}|/y_{\text{real}}^{n}$.

### E.2.2 ALIGN WITH REAL PROPAGATIONS

In the experiment, for each propagation, we set the maximum number of time steps to 50, with each time step representing 3 minutes in the sandbox. For action space, we only include like, repost, follow, and do nothing, other actions are removed to simplify the settings due to the model's limited capacity and the inadequate real-world user data we have collected. Ultimately, we would compare the simulation results for these 150 minutes with the propagation process in the real data for the first 150 minutes. For real-world time consumption, it takes 26 minutes to run a simulation that includes 300 agents for 30 time steps on one NVIDIA A100-SXM4-80GB.

Additionally, to demonstrate the reproducibility of our experiments, considering that the noise introduced by posts from other users could theoretically destabilize the propagation of the source post, we randomly select two topics: one with 33 additional posts and another with no noise. We repeat the simulation ten times for each topic and plotted the resulting curves in a single figure to illustrate the discrepancies across the ten simulations. The simulation results for the topic without noise are more stable. In contrast, the results for the other topic exhibit a divergent trend, while six out of ten experiments yield relatively concentrated results, furthermore, the degree of disturbance caused by other posts is influenced not only by the number of posts but also by the prominence of the poster. For instance, if a superuser from this group posts additional content, the propagation of the source post is likely to be affected more significantly, fortunately, this situation is rare in our dataset, and the count of additional posts is relatively small since we only consider posts created within one hour prior to the source post's creation time as noise. Overall, the simulation results are still relatively stable.

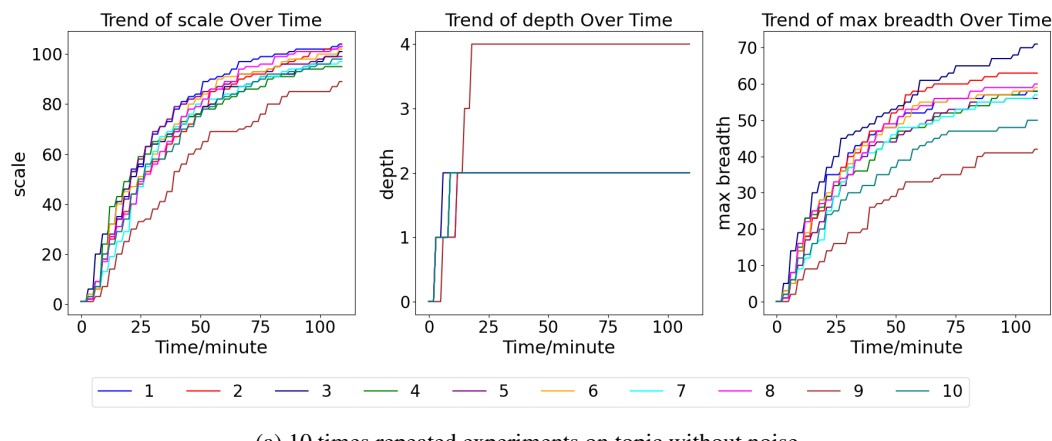

(a) 10 times repeated experiments on topic without noise.

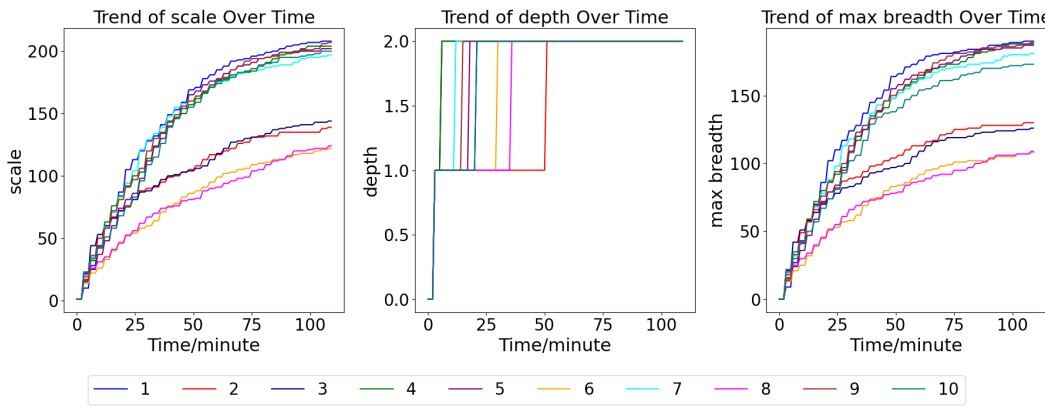

(b) 10 times repeated experiments on topic with 33 additional posts.

Figure 13: 10 times repeated experiments on 2 topics.

### E.3 GROUP POLARIZATION

#### E.3.1 DILEMMA QUESTIONS

Helen is a successful writer who usually writes popular western
    novels. Now, she has an idea for a new novel that could really
    make a big impact. If it works out, it could greatly improve
    her career. But if it fails, she will have spent a lot of time
    and effort for nothing.

What do you think Helen should do?

#### E.3.2 POLARIZATION EVALUATION PROMPTS

Please determine which of the following two sentences is more
    extreme. If there is no obvious difference in tone, return "
    neutral."

[User Question]
{question}

What do you think Helen should do?

[Answer1]

```
{answer1}

[Answer2]
{answer2}

[Response Format]
Reason:
Choice: Answer1 or Answer2 or neutral
```

### E.3.3 HELPFULLNESS EVALUATION PROMPTS

```
Please help me evaluate the helpfulness and quality of the
    responses provided by two AI assistants to the user question
    displayed below. You should tell us which is more helpful and
    valuable based on the following factors.
1. "Accurate Information": Ensure the AI provides information that
    is factual and up to date. 21
2. "Clarity and Comprehensibility": Check if the AI delivers
    information in a clear and easily understandable manner.
3. "Completeness of the Response": Ascertain that the AI answers
    all aspects of the user's query.
4. "Contextual Understanding": The AI should demonstrate a clear
    understanding of the context of the user's query.
5. "Creative Problem-Solving": If applicable, observe if the AI
    proposes creative solutions to the user's problem.
6. "Depth of Explanation": Examine whether the AI provides
    detailed and in-depth responses when required.
7. "Politeness and Professionalism": The AI should deliver
    responses using respectful and professional language.
8. "Reference to Reliable Sources": If the AI claims certain facts
    , it should be able to refer to recognized and trusted sources
    .
9. "User Engagement": The AI should engage the user effectively
    and pleasantly, encouraging positive user interaction.

[User Question]
{question}

[Answer1]
{answer1}

[Answer2]
{answer2}

[Response Format]
Reason:
Choice: Answer1 or Answer2
```

### E.4 HERD EFFECT

#### E.4.1 METRICS

We utilized two primary metrics to assess the herd effect: the post score and the disagree score. These metrics were derived from two aspects: the behavior of liking or disliking by the LLM Agent, and the content of the comments it generated.

- **Post Score**: The score ($S_i$) of a Reddit post is the difference between its upvotes ($U_i$) and downvotes ($D_i$), and can be negative:

$$S_i = U_i - D_i$$

where $S_i$ is the score of the $i^{th}$ post, $U_i$ the number of upvotes, and $D_i$ the number of downvotes.

• **Disagree Score**: In this experiment 3.4.2, where all posts were counterfactual, we examined comments from LLM agents at various time points. Using GPT-4o, we assessed how much these comments disagreed with the posts. The evaluation was guided by the following prompt template:

```
Someone posted a counterfactual statement: "{post_content}",
    and then another person commented: "{comment_content}".
Please provide a rating from 1 to 10 to indicate how much this
    person agrees with the counterfactual content, where:
1 means fully believing and agreeing with it, and 10 means
    completely disbelieving and opposing it.

Respond only with a JSON object like this:
{{
    "score": 7
}}

Ensure the "score" is a single integer between 1 and 10.
```

Before the experiment began, we randomly divided the comments data for experiment 3.3.2 and the posts data for experiment 3.4.2 into three groups (up-treated, down-treated, and control). After the experiment concluded, we calculated the mean **post score** and the 95% confidence interval of the normal distribution for all posts in the three groups of experiment 3.3.2. Similarly, we calculated the mean **disagree score** and the 95% confidence interval of the normal distribution at each time step for all comments associated with posts in the three groups of experiment 3.4.2.

### E.4.2   SETTING DETAILS

**Comparison with the Herd Effect in Humans.**   Our experiment 3.3.2 replicated the setup of a human study, including the visibility of comment scores (upvotes minus downvotes) and prohibiting the revocation of likes and dislikes, utilizing Reddit's popularity-based recommendation algorithm. To minimize biases stemming from the identities of commenters and voters and their interactions, which were meticulously accounted for in the human experiments, we manipulated a specific user to post content at scheduled intervals. This approach was adopted to mitigate the influence of different posters on the behavior of agents, and we further circumvented the impact of relationships with specific posting users on the outcomes by prohibiting agents from following or muting operations.

Consequently, the action space for the experiment included actions: like comment, dislike comment, like post, dislike post, search posts, search users, trend, refresh, and do nothing. The controlled user generated 200 posts at each time step, with each post accompanied by 1-10 comments. The recommendation system cached the top 300 posts with the highest heat scores for each agent, and each agent had a 0.1 probability of activation at every time step. Activated agents would randomly sample one of these 300 posts to read during that time step. The experiment was conducted over a total of 40 time steps.

**Herd Effect Towards Counterfactual Content.**   The action space of the experiment 3.4.2 includes create comment, like comment, dislike comment, like post, dislike post, search posts, search users, trend, refresh, and do nothing. Each agent has a 0.1 probability of activation at each time step, and each activated agent will randomly sample 5 posts from the recommended cache to read during that time step. As the number of agents increases from 100, 1k to 10k, the number of posts cached by the recommendation system respectively becomes 50, 500, and 5000. The controlled user creates 30, 300, 3k posts at each time step, respectively, until all posts in the corresponding datasets (with 219, 2191, and 21919 posts, respectively) have been created. And the experiment was conducted over a total of 30 time steps.

### E.4.3   EXAMPLES OF RESULTS

In experiment 3.4.2, 10,000 agents were able to discuss their views on counterfactual posts in the comment section, interacting by posting their own comments or by liking or disliking others' com-

ments. Over the course of the discussion, there was a gradual shift towards opposing the counterfactual content, achieving factual correction at the group level. The figure 14 below shows one such example.

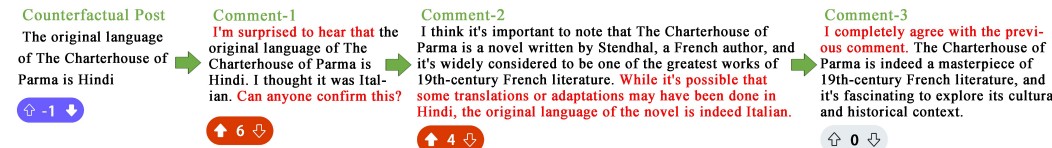

Figure 14: Example of agents' comments on counterfactual posts. As interactions increase, agents' viewpoints gradually shift from surprise and curiosity, to partial opposition, and finally to complete rejection of the counterfactual content.

## F    LIMITATIONS & FUTURE DIRECTIONS

**RecSys**    The current recommendation system is only designed at a high level similar to platforms like X (formerly Twitter) or Reddit. For example, the RecSys designed following X's model only recommends semantically similar posts based on the user's profile and recent activity. More complex recommendation algorithms, such as collaborative filtering, have not been implemented in *OASIS*, leading to a misalignment between *OASIS*'s performance and real-world propagation data.

**User Generation**    Whether we obtain user data through the Twitter API or the User Generation algorithm proposed in *OASIS*, both approaches abstract the real individual to some extent, leading to a natural gap between our simulator and the real world.

**Social Media Platform**    Although we have expanded the action space on social media platforms to a considerable extent, not all possible actions are covered. For example, our platform currently does not support features like bookmarking, tipping, purchasing, or live streaming, which could be added in future work. Additionally, the current simulation operates solely in a text-based environment, meaning agents are unable to perceive images, videos, or audio. Future extensions could incorporate multimodal content to enhance the realism of the simulation.

**Scalable Design**    While our asynchronous design helps to avoid bottlenecks, simulating millions of agents still requires several days to complete. Optimizing inference speed and improving the efficiency of database systems will be critical in reducing time and cost, making large-scale social simulations more feasible for widespread applications in the future.

**Untapped Potential**    Our large-scale social simulation platform has the potential to serve as a foundational environment for other research. For instance, it can be used to evaluate the performance of novel recommendation systems or to train large language models (LLMs) with enhanced influence capabilities, using feedback from other agents in the network as a reward signal.

## G    SOCIAL IMPACT AND ETHICAL CONSIDERATIONS

The development and application of *OASIS* provide valuable insights into complex social phenomena such as information propagation, group polarization, and herd effects. However, this also raises important ethical considerations. First, the replication of real-world social dynamics using large language model (LLM) agents introduces concerns regarding the fidelity and interpretation of the results. The risk of reinforcing biases, especially in areas related to misinformation or polarization, could exacerbate real-world issues if not properly managed. Researchers using *OASIS* must be cautious in how these simulations influence public understanding or policy recommendations.

Another key concern is privacy. While *OASIS* is designed to replicate social media environments, the use of real-world data for training agents may introduce risks related to user anonymity and

data security. Ensuring the ethical handling of any real-world datasets, including anonymization and consent, is crucial.

Lastly, the scalability of *OASIS*, while an asset for research, also presents potential dangers if misused. Large-scale agent-based models, particularly those that simulate millions of users, could be leveraged for unethical purposes such as manipulation of online discourse or misinformation campaigns. It is therefore essential to implement strict governance and ethical guidelines to prevent misuse of the simulator's capabilities.

