# OpenReview forum: "OASIS: Open Agents Social Interaction Simulations on a Large Scale"
_ICLR.cc/2025/Conference — Submitted to ICLR 2025_

### Official Review · Reviewer_HBvN · 2024-11-02

**Soundness:** 2
**Presentation:** 3
**Contribution:** 2
**Rating:** 3
**Confidence:** 5

**Summary:**

This paper approaches the problem of social simulation. The authors build about 1 million users and use LLM agents to simulate 21 kinds of actions. The authors evaluate the system by observing the trend of scale, trend of depth, trend of max breadth, etc. The results indicate that the developed system, named OASIS, can partially replicate real-world phenomena.

**Strengths:**

- The authors use real-world datasets from X and Reddit and simulate a very large-scale user community.
- There are some new features compared with the existing works, including a dynamic network and more actions.
- The paper is generally well-written.

**Weaknesses:**

- It is very strange and may introduce bias to simulate the recommendation system in a social network, which is not so important. Users can interact with information in multiple ways, and the recommendation system is only one of them. Introducing recommendation algorithms in social simulation may introduce bias as we don't know the details of recommendation algorithms in the collected data. I have also checked the papers in Table 1, in which only RecAgent and Agent4Rec considered recommendation systems (of course, as they are about the simulation of recommendation rather than social simulation).
- The evaluation should be improved. Can we use the results in Figure 4 to claim that the simulation is accurate and useful? There are still errors. The authors should carefully analyze where the error comes from and help the readers know when we can use the results of OASIS. In other words, what kind of use can these imperfect simulations have? We all know that perfect simulations are not possible, but it is important to analyze what kind of impact these errors may have.
- The authors should conduct more evaluations based on social network theory. Could the simulation system be consistent with those representative theories such as network dynamics? These important and basic questions have not been well answered.

**Questions:**

The questions are based on the weaknesses, and it will be great if the authors can offer feedback.
1. Why simulate recommendation system? What kind of bias it involves?
2. Can we use the results in Figure 4 to claim that the simulation is accurate and useful? What kind of use can these imperfect simulations have?
3. Could the simulation system be consistent with those representative theories such as network dynamics?

**Details Of Ethics Concerns:**

The social simulation may involve two kinds of ethical issues.
1. The privacy issue in data collection and usage.
2. The potential risk if the LLM agents are used to spread misinformation, harmful messages, etc.

---

> ### Author Response · Authors · 2024-12-04
>
> **[w1]** It is very strange and may introduce bias to simulate the recommendation system in a social network, which is not so important. Users can interact with information in multiple ways, and the recommendation system is only one of them. Introducing recommendation algorithms in social simulation may introduce bias as we don't know the details of recommendation algorithms in the collected data. I have also checked the papers in Table 1, in which only RecAgent and Agent4Rec considered recommendation systems (of course, as they are about the simulation of recommendation rather than social simulation).
>
> Thank you for your question. Our recommendation system is built based on the official open-source framework provided by Twitter. The recommendations are categorized into 'in-network' and 'out-network.' In-network recommendations are based on users' following relationships, where agents can follow users of interest to receive more tweets from them in the future. Out-network recommendations, on the other hand, are driven by interest-based suggestions, where tweets that match an agent's interests are recommended. Therefore, a recommendation system is essential for platforms like Twitter. We use Twhin-BERT for our recommendation system, which has been pre-trained on a large amount of Twitter data, enabling it to model user interests effectively and provide better interest-based recommendations. Regarding bias, we believe that our recommendation system, like those of existing social media platforms, may lead to the creation of information bubbles, where agents are predominantly exposed to information that aligns with their interests. We will further address this issue of bias in future versions.
>
> **[w2]** The evaluation should be improved. Can we use the results in Figure 4 to claim that the simulation is accurate and useful? There are still errors. The authors should carefully analyze where the error comes from and help the readers know when we can use the results of OASIS. In other words, what kind of use can these imperfect simulations have? We all know that perfect simulations are not possible, but it is important to analyze what kind of impact these errors may have.
>
> Thank you for your question. In our message propagation experiments, we compared the message propagation process in OASIS with that in the real world. We found that while the broadcast mechanism in OASIS aligns well with real-world behavior, there is a significant discrepancy in the depth of the propagation paths. We provide an explanation for these differences in Section 3.3.1 of the paper. Understanding these discrepancies can help users better tailor their use cases and further improve the capabilities of OASIS.
>
> **[w3]** The authors should conduct more evaluations based on social network theory. Could the simulation system be consistent with those representative theories such as network dynamics? These important and basic questions have not been well answered.
>
> Thank you for your question. In our message propagation experiments, we compared the propagation paths with those observed in the real world. In future versions, we plan to include additional comparisons related to social network dynamics, such as the Schelling segregation model.

---

### Official Review · Reviewer_AvZ8 · 2024-11-04

**Soundness:** 3
**Presentation:** 3
**Contribution:** 3
**Rating:** 8
**Confidence:** 3

**Summary:**

This manuscript proposes an LLM-based ABM simulation framework, OASIS, to study large-scale social phenomena arising from social interactions in a multi-agent social media environment. Compared with existing frameworks, OASIS can host many users (1M), simulate X & reddit, have a bigger action space, implement recommendation systems, and simulate dynamic network evolution. The authors demonstrate the utility of OASIS by simulating several social phenomena such as information spreading, group polarization, and herding behavior.

**Strengths:**

- The OASIS framework seems to be an impressive and comprehensive piece of engineering for simulating large-scale social media environments with LLMs, which can be useful for future studies on large-scale social phenomena.

- The framework seems to be capable of exhibiting somewhat realistic behavior in several social phenomena, which is a promising sign.

- Many ablation studies are performed to show the robustness of the results and the importance of each component of the framework.

- The paper is fairly well-written and relatively easy to follow.

**Weaknesses:**

- I think, in a way, the paper tries to pack too much in a single paper. I think it was challenging to provide all the details of the engineering AND conduct thorough social experiments. As a result, I feel that the social phenomena simulations are somewhat shallow and not very insightful. For instance, the simulations raise the question of "null models" -- how surprising or insightful are these massive LLM-simulated results compared to a simple model? -- as well as the question of potential parameter and environment settings. Given so many parameters one should set to simulate a social media, it is really hard to know what is driving the results and how different the simulation results would be in different settings. It is simply impossible to know whether the results shown in the paper are typical across different settings or something that only arises in the specific settings used in the paper. While I think it is still meaningful to demonstrate that the OASIS framework can produce somewhat realistic social phenomena, I think it could have been more insightful if there was a separate, in-depth analysis of each social phenomena.

- I was not entirely sure whether the paper contributes to the advancement of representation learning. The paper uses LLMs, but the focus is more on the engineering of the simulation framework. Although it may be possible to argue that the paper enables the study of large-scale social phenomena that can be studied with LLMs and how the "representation" that LLMs use can affect the social phenomena, it is still quite indirect.

**Questions:**

- Studies have shown that many large-scale patterns from social media can be explained by simple models and few key mechanisms. For instance, Weng et al. Sci. Rep. (2012) showed that limited attention is enough to produce the power-law distribution of propularity of hastags; the "burstiness" of human behavior (see Barabasi's work on burstiness) has been studied as a key mechanism for many patterns in social phenomena; the "homophily" mechanism has been shown to be a key mechanism for many social phenomena; There are also many cognitive biases that can explain many social phenomena.

I don't think it is required for this paper at this stage to incorporate all these mechanisms into the OASIS framework, but I think it would be great to discuss how the OASIS framework can be used to study these mechanisms and how the results from OASIS can be interpreted in light of these cognitive mechanisms.

---

> ### Author Response · Authors · 2024-12-04
>
> **[w1]** I think, in a way, the paper tries to pack too much in a single paper. I think it was challenging to provide all the details of the engineering AND conduct thorough social experiments. As a result, I feel that the social phenomena simulations are somewhat shallow and not very insightful. For instance, the simulations raise the question of "null models" -- how surprising or insightful are these massive LLM-simulated results compared to a simple model? -- as well as the question of potential parameter and environment settings. Given so many parameters one should set to simulate a social media, it is really hard to know what is driving the results and how different the simulation results would be in different settings. It is simply impossible to know whether the results shown in the paper are typical across different settings or something that only arises in the specific settings used in the paper. While I think it is still meaningful to demonstrate that the OASIS framework can produce somewhat realistic social phenomena, I think it could have been more insightful if there was a separate, in-depth analysis of each social phenomena.
>
> Thank you for your valuable suggestions. The experimental settings for our three social scenarios align with those used in social science literature. At this stage, our goal is to replicate the phenomena observed in these classic sociological experiments in order to validate the effectiveness of OASIS across different settings. In future versions, we plan to incorporate deeper analysis of each scenario.
> Regarding the issue of parameter control influencing results that you mentioned, I believe your thoughts are valuable, and we have also considered this. In our paper, when setting up the platform, our consideration in parameter setting was to align as closely as possible with real social media, not to manipulate the results. We also tried to avoid introducing decision bias when setting agent prompts. The reason for this setup is to faithfully explore the similarities and differences between agent societies and human societies.
> Furthermore, while the possibility of using a single model for group simulations is an interesting direction, we believe that the multi-agent interaction approach currently offers greater flexibility and interpretability. This approach allows us to more easily configure settings and analyze the states of individual agents.
>
> **[w2]** I was not entirely sure whether the paper contributes to the advancement of representation learning. The paper uses LLMs, but the focus is more on the engineering of the simulation framework. Although it may be possible to argue that the paper enables the study of large-scale social phenomena that can be studied with LLMs and how the "representation" that LLMs use can affect the social phenomena, it is still quite indirect.
>
> Thank you for your valuable suggestions. We believe that understanding the representations of LLMs is crucial for studying the collective behavior of LLM-based agents. Collective behavior not only helps us assess the capabilities of individual agents, but also enhances their performance through interactions within the group.
>
> **[Q1]** Studies have shown that many large-scale patterns from social media can be explained by simple models and few key mechanisms. For instance, Weng et al. Sci. Rep. (2012) showed that limited attention is enough to produce the power-law distribution of propularity of hastags; the "burstiness" of human behavior (see Barabasi's work on burstiness) has been studied as a key mechanism for many patterns in social phenomena; the "homophily" mechanism has been shown to be a key mechanism for many social phenomena; There are also many cognitive biases that can explain many social phenomena
>
> Thank you for your insightful suggestions. We will conduct further research in future versions to investigate whether the aforementioned characteristics exist within agent-based societies.

---

### Official Review · Reviewer_ZN1L · 2024-11-04

**Soundness:** 3
**Presentation:** 3
**Contribution:** 2
**Rating:** 5
**Confidence:** 5

**Summary:**

This paper introduces OASIS, a scalable and generalizable social media simulator designed for studying complex systems with large language model (LLM) agents. Unlike existing simulators, which are scenario-specific and limited to small agent counts, OASIS can simulate up to one million users on platforms like X and Reddit, incorporating dynamic environments, diverse actions (e.g., following, commenting), and adaptive recommendation systems. OASIS replicates social phenomena such as information spreading, group polarization, and herd effects across various platforms. Results show that larger agent groups enhance group dynamics and yield more diverse opinions, demonstrating OASIS’s versatility for large-scale digital social studies.

**Strengths:**

1. I am very interested in the topic of this paper; it is a large-scale social simulation that exceeds the number of agents used in current social simulations.

2. This paper presents a thorough piece of work with a substantial engineering effort. The meticulous design of each module contributes to excellent results.

3. This paper excels in writing, experimental illustration, and analysis, providing a great experience for readers.

4. The topic of this paper is very meaningful; social simulation is highly important and crucial for the development of social media and computational social science.

**Weaknesses:**

1. The primary contribution of this paper is significantly increasing the scale of existing social simulations to the one-million level. However, I did not find scientific insights on this issue in the paper, such as:

* No explanation of why previous methods could not scale to the million level, is there any novel algorithms, data structures, or distributed computing techniques used?
* No details on the methods used to overcome this bottleneck

The approach in this paper seems more of an engineering solution rather than a scientific discovery, which does not sufficiently support the argument presented in the paper. The authors include a more in-depth comparison to previous approaches, highlighting the key differences that allowed for increased scale. This would help clarify whether there are scientific contributions beyond engineering optimizations.

2. An important contribution of this paper is incorporating dynamically updated environments, diverse action spaces, and recommendation systems. The diverse action spaces are a good innovation, aligning well with the actions on social media platforms in real life. However, I have some questions regarding the dynamically updated environments and recommendation systems:

* For dynamically updated environments, I understand this should involve continuous changes in users or relationships between users.  I think the authors should tell us how new users are added, how relationships evolve over time, or how the content of the simulated social media platform changes as agents interact.

* Many social simulations already include a recommendation systems module[1], so as a contribution to this paper, the recommendation systems module lacks novelty.

3. This work lacks comparisons with previous methods. For example, in the Information Propagation experiment, previous studies have also explored similar experiments[2]. The paper should include some comparison results to demonstrate the advantages of OASIS.  (e.g. scalability, accuracy of replicating real-world phenomena, computational efficiency). (I am not necessarily requesting that these experimental results be provided at the rebuttal stage, nor do the baseline selections need to be limited to my references. However, I believe that before this paper is published, it should include the results of this comparison experiment.)

4. In section 3.4, this part explores "DOES THE NUMBER OF AGENTS AFFECT THE ACCURACY OF SIMULATING GROUP BEHAVIOR?" However, in the two experiments (“INFORMATION PROPAGATION IN X” and “HERD EFFECT IN REDDIT”), there is no detailed explanation of how accuracy is defined for each experiment.

5. This paper does not provide data and code, which raises significant concerns about its reproducibility.

6. There is a typo on line 411, “curs?”.


References：

[1]. Wang L, Zhang J, Yang H, et al. User behavior simulation with large language model based agents[J]. arXiv preprint arXiv:2306.02552, 2023.

[2]. Mou X, Wei Z, Huang X. Unveiling the truth and facilitating change: Towards agent-based large-scale social movement simulation[J]. arXiv preprint arXiv:2402.16333, 2024.

***


For the above reasons, I believe this paper still requires more time and further work to be refined. However, it is very meaningful, and the topic is highly interesting.

**Questions:**

First, I hope the authors can respond to the points outlined in the weaknesses section.

Then, I have some additional questions I would like to discuss with the authors.

1. On line 180, the paper mentions a "relations network." In what format is this relations network stored? Is it in the form of a graph, text, or something else?

2. On line 341, the paper mentions a "propagation graph." In what format is this propagation graph stored? Is it in the form of a graph, text, or something else?

3. How is an “uncensored LLM” defined?

4. For Comment scores, why is it calculated as upvotes minus downvotes? If a post has 100 upvotes and 100 downvotes, while another post has 5 upvotes and 5 downvotes, both would have a Comment score of 0. However, the level of attention these two posts receive on social media is clearly different.

5. Why not use models like GPT-3.5-turbo, GPT-4 (possibly more costly), or GPT-4o-mini for the experiments? Given their relatively strong reasoning abilities, they might achieve higher accuracy.

---

> ### Author Response · Authors · 2024-12-04
>
> **[w1]** The primary contribution of this paper is significantly increasing the scale of existing social simulations to the one-million level. ...
>
> *Challenge 1: Inference Scheduling.* We manage the immense volume of inference requests from one million agents using a system of asynchronous requests and concurrent processing through multithreading. This allows for efficient simulation of large agent numbers.
>
> *Challenge 2: Large-Scale Agent Interaction.* To handle the complex information flow among agents, we use a recommendation system based on interest and hot-score recommendations, simplifying large-scale simulations. Future work will refine these solutions and compare them with other methods.
> We will further refine these solutions and provide a comparative analysis with alternative methods in the paper.
>
> **[w2]** An important contribution of this paper is incorporating dynamically updated environments, diverse action spaces, and recommendation systems. ...
>
> Thank you for your insightful question. Firstly, on dynamic environments: Agents are initialized from real-world data and interact similarly to real-world social media platforms. They receive post recommendations, decide actions based on a large language model, and dynamically update relationships and the post database through various actions.
>
> Secondly, on the recommendation system: Traditional recommendation systems, like Light GCN, are less suitable for dynamic social media platforms like Twitter, where content is constantly evolving. To our knowledge, no research has specifically designed recommendation systems for such platforms.
>
> **[w3]** This work lacks comparisons with previous methods. ...
>
> Thank you for your question. While previous works have indeed addressed message and rumor propagation, the differences in the social simulation settings make it challenging to directly align our work with others. As a result, a fair comparison with existing studies is difficult to achieve. In future versions of this work, we plan to dedicate more effort to benchmarking and comparing our approach with other relevant research.
>
> **[w4]** In section 3.4, this part explores "DOES THE NUMBER OF AGENTS AFFECT THE ACCURACY OF SIMULATING GROUP BEHAVIOR?"
>
> Thank you for your question. In the herd behavior experiment, the accuracy of the results is assessed by the confidence intervals of the disagree score, as shown by the differently colored regions (red, green, and blue) in Figure 8. Comparing the confidence intervals, rather than just the means, helps distinguish between results caused by random fluctuations and those that exhibit statistical significance. This approach is similar to the concept of increasing sample size in sociological surveys. We observed that when the number of agents was 100, the confidence intervals for the three groups were large and overlapped, indicating that the results might be unreliable. However, when the number of agents increased to 10,000, the confidence intervals narrowed, and the down-treated group clearly separated from the other two groups, providing strong evidence for the accuracy of the experiment.
>
> **[w5]** On line 180, the paper mentions a "relations network." In what format is this relations network stored? ...
>
> Regarding the 'relations network,' we store the edges of the social network in a relational database as part of the overall social media platform database. See follow table in Apendix C.2.
>
> The basic unit of the propagation path is the post-to-post forwarding relationship. By tracking the post-to-post forwarding graph, we can calculate the depth and breadth of propagation. The agent-to-agent forwarding relationships are recorded in the Trace table of the database.
>
> **【w6】** 1. For Comment scores, why is it calculated as upvotes minus downvotes? ...
>
> It is correct that many social media platforms display both the number of upvotes and downvotes on posts. However, Reddit follows a different approach, as it only shows the score(see https://www.reddit.com/). A study on herd behavior in humans, published in Science (https://www.science.org/doi/10.1126/science.1240466), conducted experiments on a platform similar to Reddit, where only scores are displayed. We have replicated their experimental setup in our own research.
>
> **【w7】** 1. How is an “uncensored LLM” defined?
>
> The term 'uncensored LLM' refers to a series of LLMs that have had their safety barriers removed, significantly compromising their security. For more details, refer to [https://huggingface.co/blog/mlabonne/abliteration].
>
> **【w8】** 1. Why not use models like GPT-3.5-turbo, GPT-4 (possibly more costly), or GPT-4o-mini for the experiments? Given their relatively strong reasoning abilities, they might achieve higher accuracy.
>
> Due to the large-scale user simulation we conduct, running a full experiment incurs significant costs. In future versions, we plan to incorporate powerful proprietary models to conduct large-scale message propagation experiments.

---

### Official Review · Reviewer_eewh · 2024-11-06

**Soundness:** 2
**Presentation:** 2
**Contribution:** 1
**Rating:** 1
**Confidence:** 2

**Summary:**

In this paper, the authors present an LLM-based platform for the agent-based simulation of user behavior and interactions in online social media at large scale. The system consist of multiple components, which allow to incorporate data on real-world users on social media platforms and generate a large number of artificial agents by prompting a large language model. A time module controls the temporal acticity of those agents and an LLM is used to generate actions such as the creation of new posts, liking or disliking the posts of others or forming new follower relationships. A recommender system module that models recommendation algorithms of real platforms determines which content is shown to agents. The authors include experimental results on simulations of information spreading and group polarization on the platform X, and herd effects in terms of comment scores on the platform Reddit. Aggregate scores are used to compare the results of these simulations to behavior of humans.

**Strengths:**

[S1] The work highlights an interesting direction of research on leveraging large language models for realistic agent-based simulations of collective social behavior at large scale.

[S2] The authors present a practical design of a platform for agent-based simulations at large scale.

[S3] The system is experimentally evaluated by comparing the results of agent-based simulations of information spreading and group polarization with actual human behavior.

**Weaknesses:**

[W1] Given the complexity of LLMs, it is unclear which insights an LLM-based agent-based simulation can yield into the actual microscopic mechanism underlying collective social phenomena that emerge at the macroscopic scale.

[W2] The description of the platform remains at a rather high and superficial level. Some details of the complex platform design are not explained in detail.

[W3] I am not convinced by the way how temporal and topological characteristics of agent interactions are integrated into the simulation.

[W4] I do not think that the contributions summarized in this paper are of sufficient relevance for the wider ICLR research community.

**Questions:**

Regarding [W1], the authors clearly position their work as a tool for researchers who are interested to investigate complex collective phenomena emerging in social systems at large scale. To that end, agent-based models are typically employed in computational social science not to faithfully simulate all aspects of a given system but to understand how simple mechanisms of agent behavior and agent interactions at the micro-scale give rise to collective phenomena at the macro-scale. A crucial aspect in this approach is thus the ability to trace back a given phenomenon at the macro-scale to a specific mechanism operating at the microscale between individual agents. Given the approach proposed in this work and the complexity of the LLM models governing agent dynamics, it is unclear how the design of the system can help to gain insights into such mechanisms. Could the authors comment on this?

Referring to [W2]. from the rather short descriptions of the individual components, I did not understand many crucial details of the platforms and the experiments. For instance, what exact data on real users are used to initialize the simulation and how much of such data is needed. How have the specific users in the experimental evaluation (e.g. for the group polarization experiment) been chosen? How were the features of users (e.g. profession, Myers–Briggs Type Indicator) chosen and which data were used to fit the distribution? How exactly was the iniital social network between agents generated and how realistic is this compared to relationships between real users? What is the rationale behind the specific prompts used to generate agents or actions and how could the formulation of the prompt change the result? Related to [W1] all of those aspects could be thought as "parameters" in an immensely complex agent-based simulation, which may crucially influence the results!

Regarding [W3], it is well-known in the network science and complex systems community that both the topology of interactions between agents as well as their temporal dynamics crucially influences collective dynamics in multi-agent systems. While both aspects are (somehow) considered by the Time Engine and the initial network generated when agents are created (and new follower relations generated by the action module), it is unclear how realistic those two crucial components are. To be more specific, a simple vector of hourly probabilities of activity underfits the real complexity of human dynamics on social media, as it assumed a memoryless process. In contrast, real-world human behavior on social media has been shown to be characterized by complex non-Markovian and bursty activity patterns which are not captured by this approach. Similarly, the initial follower network will likely strongly influence agent behavior as well as the evolution of agent relations. It is not clear to me how realistic the proposed approach for network initialization is.

Finally and considering [W4], I am not convinced that the contributions made by this work are sufficient to be relevant for the ICLR research community. As also clearly stated by the authors, the general idea to use LLMs for more realistic agent-based simulations is not new and has been explored in several recent works. One key contribution of this work is that it proposes a practical design how such LLM-based agents can be integrated with a rule-based system to facilitate large-scale simulations of agent societies.

The fact that the developed system has been evaluated experimentally against a population of human users is noteworthy, but I am not convinced that the rather simple aggregate results (e.g. on the scale, breath, and depth of information propagation paths or on simple mean comment scores) actually support the strong claim that the proposed approach allows to replicate real-world social phenomena. Moreover, to justify the considerable computational resources required to perform LLM-based agent simulations at large scale, it would be necessary to compare those results to simpler agent- or network-based models for information propagation or opinion formation.

In summary, I am not convinced that the contributions of this work justify publication at ICLR. To me, this work seems more fitting to a venue focussing on agent-based simulations, simulations of social dynamics, computational social science, or so-called "agent societies". My low score on the contribution of this work - and the recommendation to reject it - should thus be interpreted in the context of a deep learning venue and does not imply that this work does not make a contribution that is more valuable for other communities.

**Details Of Ethics Concerns:**

An important ethical concern is that variants of the proposed approach could be used to create large numbers of fake users, which by realistically simulating human behaviour could help to spread manipulated information and influence opinions at large scale in a potentially harmful way.

I do not - by any means - suggest that this is the intention of the authors, but I still believe that it is a legitimate concern. This has actually been correctly mentioned as a potential ethical concern by the authors in the appendix, which is why I deem it necessary to also highlight it here.

---

> ### Author Response · Authors · 2024-12-04
>
> **[w1]** Given the complexity of LLMs, it is unclear which insights an LLM-based agent-based simulation can yield into the actual microscopic mechanism underlying collective social phenomena that emerge at the macroscopic scale.
>
> Thanks for the reviewer’s question. The main confusion raised by the reviewer is at the micro and macro levels of analysis. OASIS, in its simulation process, records the motivation behind each agent’s actions, which is done through the COT technique. This allows us to track the fundamental behaviors of the agents. For example, our statistical analysis reveals that when agents encounter rumors, they tend to use emotional language such as "interesting" or "attractive," indicating that the rumors trigger the agents' emotions. These analyses help us to understand the underlying causes of group behaviors, which we have discussed thoroughly in the paper.
>
> Moreover, the design of our platform aids in replicating the complexity of interaction dynamics in real social media platforms and maintains versatility for various social experiments. For instance, the number of likes on a post, the content of comments, and the recommendation system all interact with each other, mirroring the way real human social media information dissemination is influenced by these interconnected factors.
>
> Many studies have demonstrated the ability of LLM agents to simulate human behavior[1][2], affirming the feasibility of using LLM agents for social simulation. We understand your focus on making agents more similar to humans for better simulation, but there might be a slight misunderstanding regarding our contribution. We want to emphasize that as an initial exploratory work on large-scale social simulation with LLM agents, our aim is not to make LLM agents' behavior identical to human behavior. Instead, we are exploring the similarities and differences between agent societies and human societies, investigating the potential of agent societies in predicting human society and other aspects, or applying it to other areas such as gaming.
>
> [1] Social simulacra
>
> [2] SOTOPIA
>
> **[w2]** The description of the platform remains at a rather high and superficial level. Some details of the complex platform design are not explained in detail.
>
> Thank you for the reviewer’s question. In Section 3.2, line 309, we describe how we collect real user data and the main data we use for the simulation. In Section 3.2, line 312, we explain that the users for the group polarization experiment are sourced from the real users involved in the message propagation experiment. For the generated users, we provide a detailed discussion in Section 2.6 and Appendix D.2. As for the prompt, it mainly consists of two parts: the user description and the action generation. Due to space limitations, we did not conduct ablation studies on the specific details of the prompt. We will conduct more ablation in the future.
>
> **[w3]** I am not convinced by the way how temporal and topological characteristics of agent interactions are integrated into the simulation.
>
> Thank you for your insightful comments. User time behavior is challenging to model due to its multifactorial nature; we use a simple activation probability model to address this. It approximates user time behavior and scales well for large agent simulations. We aim to improve temporal feature modeling in the future. As for the social network, we classify users into core and regular users, based on prior work. This method aligns with Twitter's network structure. More details are in Appendix D.2.
>
> **[w4]** I do not think that the contributions summarized in this paper are of sufficient relevance for the wider ICLR research community.
>
> Our innovation primarily lies in constructing a realistic, universal social media environment for studying large-scale agent interactions. Past work in this field has often focused on specific scenarios and oversimplified many platform features. In contrast, with the increasing capabilities of large models and agents, our work has built a highly realistic LLM agent social simulation platform based on real platform features, making a significant contribution as an agent infrastructure. We have put substantial effort into the engineering aspect, and the results not only facilitate the social simulation experiments we've mentioned but also provide a platform for other researchers to conduct a wider range of studies on subjects such as LLM, data generation, recommendation systems, and more.
>
> **Ethics Concerns**
> Our experiments are conducted in a controlled environment, without the involvement of real human participants. We believe that our platform can serve as a valuable tool for advancing research on LLM safety, offering insights into emerging security challenges in agent-based societies. By providing a realistic simulation of agent interactions, our platform aims to guide the development of strategies to address potential safety risks in future agent-driven ecosystems.

---

### Meta-Review · Area_Chair_vwV9 · 2024-12-23

**Metareview:**

The paper introduces a (scalable) LLM-based social media simulator and demonstrate empirically that the simulator can reproduce several social phenomena such as information spreading, group polarization, and herding behavior. In terms of overall rating, two of the reviewers are pretty negative, one is mildly negative and one is positive. The main concerns comprise the motivation, contextualization with the state of the art, significance/fit of the contributions to ICLR, and experimental evaluation. In this regard, even the most positive reviewer questions the significance of the methodological contribution and the rebuttal did not mild the reviewers' concerns. In addition, two reviewers did raise ethical concerns, which I do share. As a consequence, I am unable to recommend acceptance.

**Additional Comments On Reviewer Discussion:**

The rebuttal provided by the authors did not persuade the reviewers to follow-up nor change their overall scores.

---

### Decision · Program_Chairs · 2025-01-22

Reject